# *Starship* giant transposons dominate plastic genomic regions in a fungal plant pathogen and drive virulence evolution

Yukiyo Sato [1], Roos Bex [2], Grardy C. M. van den Berg[3], Parthasarathy Santhanam[4], Monica Höfte [2], Michael F. Seidl [5] & Bart P.H.J. Thomma [1] ✉

*Starships* form a recently discovered superfamily of giant transposons in Pezizomycotina fungi, implicated in mediating horizontal transfer of diverse cargo genes between fungal genomes. Their elusive nature has long obscured their significance, and their impact on genome evolution remains poorly understood. Here, we reveal a surprising abundance and diversity of *Starships* in the phytopathogenic fungus *Verticillium dahliae*. Remarkably, *Starships* dominate the plastic genomic compartments involved in host colonization, carry multiple virulence-associated genes, and exhibit genetic and epigenetic characteristics associated with adaptive genome evolution. Phylogenetic analyses suggest extensive horizontal transfer of *Starships* between *Verticillium* species and, strikingly, from distantly related *Fusarium* fungi. Finally, homology searches and phylogenetic analyses suggest that a *Starship* contributed to de novo virulence gene formation. Our findings illuminate the profound influence of *Starship* dynamics on fungal genome evolution and the development of virulence.

Transposable elements (TEs, transposons) are ubiquitous mobile genetic elements in all life forms. Originally, these have been seen as selfish elements carrying only genetic information for their proliferation, but presently are appreciated to shape genome structure and function, and drive evolutionary innovations[1]. Whereas most TE superfamilies have simple structures and encode few proteins[2], giant TEs are tens to hundreds of kilobases (kb) in size and carry tens to hundreds of cargo genes[3–5].

*Starships* are giant TEs (15–700 kb) that were recently discovered in Pezizomycotina fungi, the largest subdivision of Ascomycota, and typically contain a tyrosine recombinase (YR) "captain" gene as the first gene, required for transposition, while cargo genes are variable and functionally diverse[5–8]. How *Starships* impact global genome evolution, and the range and extent to which *Starships* have shaped genomes

over time, remains enigmatic. *Starship* detection is technically challenging owing to the large diversity in cargo that can include abundant repeats[7], and relies on the detection of the presence/absence of YR-containing inserts in orthologous sites among highly contiguous genome assemblies[5]. While 143 *Starships* were identified in this manner in a systematic search of 2899 fungal genomes, 10,628 "orphan" captain-like YR genes remained, suggesting many overlooked *Starships*[6]. *Starships* occupy up to 2.4% of the genome of the human pathogen *Aspergillus fumigatus*, show extensive presence/absence variation, and contain many differentially expressed cargo genes upon infection[9]. Moreover, in the plant pathogen *Macrophomina phaseolina*, 30% of chromosomal translocations, inversions, and putative chromosomal fusions occur near *Starship* insertions[7]. Interestingly, *Starships* can transfer horizontally between closely-related fungi of the

[1]Institute for Plant Sciences, Department of Biology, University of Cologne, Cologne, Germany. [2]Laboratory of Phytopathology, Department of Plants and Crops, Faculty of Bioscience Engineering, Ghent University, Ghent 9000, Belgium. [3]Laboratory of Phytopathology, Wageningen University and Research, Droevendaalsesteeg 1, 6708PB Wageningen, the Netherlands. [4]Agriculture and Agri-Food Canada, Morden Research and Development Centre, Morden, MB, Canada. [5]Theoretical Biology & Bioinformatics Group, Department of Biology, Utrecht University, Utrecht, the Netherlands. ✉e-mail: bthomma@uni-koeln.de

same order, and can transfer important traits such as pathogenicity[8,10–16].

Pathogens and their hosts typically engage in molecular arms races, with the pathogen exploiting secreted virulence factors (effectors) to mediate host colonization, while hosts employ immune receptors for pathogen interception[17–19]. To avoid recognition, pathogen effector catalogs are highly dynamic and variable, mediated by a "two-speed genome" organization in which virulence genes co-localize in highly plastic genomic regions that are enriched in repetitive elements and particular epigenetic features[20–26]. Accordingly, the fungus *Verticillium dahliae* that causes vascular wilt disease in hundreds of hosts[27–29] contains plastic 'adaptive genomic regions' (AGRs)[30] that are enriched in virulence genes[31–36], transcriptionally active TEs[30,37,38], and structural variations[32,37–39], associated with a unique chromatin profile and physical interactions in the nucleus[30,40–42]. Besides *V. dahliae*, the Pezizomycotina *Verticillium* genus contains nine additional plant-associated species[28]. Thus far, two *Starships* have been identified in a single strain of *V. dahliae*[6]. Here, we queried 56 highly contiguous *Verticillium* genome assemblies for *Starships* to analyze their association with the evolution of AGRs and virulence on plant hosts.

## Results

### A wealth of *Starships* occurs in the *Verticillium* genus

To identify *Starships* across the *Verticillium* genus, we collected 56 high-quality *Verticillium* genome assemblies, comprising 36 *V. dahliae* strains and 20 strains of the nine remaining species (Fig. 1a and Supplementary Data 1, 2) and queried these genomes for *Starships* using "Starfish"[6]. We uncovered 54 *Starships* that belong to 24 haplotypes of 14 naves of seven families (Fig. 1a, b and Supplementary Data 3). Between one and three *Starships* were detected in 33 of the 56 strains belonging to seven of the ten *Verticillium* species (Fig. 1a, b). Thus, most *Verticillium* genomes contain a *Starship*, and several genomes even contain multiple. Moreover, as these *Starships* range from 17 to 625 kilobases (kb) (Fig. 1c), and larger ones typically contain multiple YR genes of different naves (Fig. 1c, d), these likely represent nested *Starship* insertions. Thus, the final number of *Verticillium Starships* is likely under-estimated.

*Verticillium* genomes exhibit extensive large-scale genomic rearrangements that may have affected the integrity of prior inserted *Starships*[32,37,43]. However, Starfish cannot identify fragmented *Starships*, rearranged *Starship* insertion sites, or *Starship* insertions into lineage-specific regions[6]. To identify such *Starships*, we queried for reference captain and captain-like YRs previously identified in Pezizomycotina genomes[6], revealing 2–18 homologs in each strain, amounting to 556 YR genes of 38 naves and seven families (Supplementary Fig. S1 and Supplementary Data 4). Only 21% of them belong to *Starships* identified by Starfish (Fig. 1e), while the remaining 79% could point to unidentified *Starships*. Additionally, to detect potential *Starships* and *Starship*-like regions that may have been overlooked due to genomic rearrangements and insertion into lineage-specific regions, we identified regions syntenic to *Starships* as "*Starship* regions". Hereafter, "*Starships*" refer to those identified using Starfish, while "*Starship* regions" collectively refer to *Starships* regions plus syntenic regions. Intriguingly, such *Starship* regions occur in all strains (Supplementary Data 5), and half of the captain-like YR genes that did not occur in *Starships* appeared in such *Starship* regions (Fig. 1e). Thus, we reveal abundant *Starships* and their remnants in the *Verticillium* genus.

### *Starships* are hotspots of large-scale genomic rearrangements

To detail how genomic rearrangements affected *Starships*, we compared telomere-to-telomere genome assemblies of *V. dahliae* strains JR2 and VdLs17 that comprise dozens of large-scale genomic rearrangements[32,37,44]. In strain JR2, we detected three large *Starships* of 0.50–0.54 Mb each, belonging to haplotypes *Ar1h1*, *Ar4h1*, and *Se1h1*, plus additional *Starship* regions, collectively accounting for

5.3% (1.92 Mb) of the genome (36.15 Mb) (Fig. 2a). Although no *Starships* were detected in VdLs17 by Starfish, *Starship* regions account for 2.7% (0.96 Mb) of the genome (35.97 Mb) (Fig. 2a). Intriguingly, 60% of the inter-chromosomal rearrangement breakpoints between these strains occurred in *Starship* regions (Fig. 2a and Supplementary Fig. S2). Accordingly, *Starship* regions were mainly detected in AGRs that are enriched in such rearrangements[32,37] (Fig. 2a, b). In strain JR2, 92% (1.77 Mb) of the *Starship* regions colocalized with AGRs (Supplementary Data 6). Moreover, 53% of the total AGR complement belongs to *Starships*. In VdLs17, 94% (0.90 Mb) of the *Starship* regions colocalized with 22% of the AGR complement (Supplementary Data 6).

Since significant AGR proportions could not be assigned to *Starships* in JR2 (47%) and VdLs17 (78%), we determined whether *Starship* regions in AGRs display different characteristics than other AGRs. Pairwise genomic alignments of the JR2 genome with those of the other 35 *V. dahliae* strains revealed significantly lower alignment coverage in *Starship* regions (median 31%) than in other AGRs (median 85%) and core genomic regions (median 93%) (Fig. 2c and Supplementary Fig. S3), indicating enhanced presence/absence variation or sequence divergence in *Starship* regions. Moreover, we revealed an extreme enrichment in structural variation, particularly concerning translocations, inversions, and duplications, in *Starships* (Fig. 2d). Regions lacking alignment accompanying these rearrangements were significantly larger in *Starship* regions (median 14.7 kb) than in other AGRs (median 5.9 kb) and core regions (median 1.7 kb) (Fig. 2e and Supplementary Fig. S4a), underscoring the extensive presence/absence variation in *Starship* regions. Intriguingly, some genomic rearrangements even occurred between *Starships* as the *Ar1h1 Starship* in *V. dahliae* strain GF1192 is largely syntenic to the JR2 *Ar1h1 Starship*, but synteny lacks at four sites that are syntenic to parts of the JR2 *Ar4h1 Starship* (Fig. 2f and Supplementary Fig. S4b), leading to *Starship* diversification.

### *Starships* display typical traits of plastic genomic regions

Like plastic genomic regions of many filamentous plant pathogens[20–24], *V. dahliae* AGRs are enriched in effector genes, *in planta* induced genes with facultative heterochromatic histone modifications (H3K27me3)[30,32,41], and repetitive elements including active TEs[37,38]. Intriguingly, the previously characterized effector genes *Ave1*[31,33] and *Av2*[34,45] occur in *Ar1h1* and *Ar4h1 Starships* (Fig. 3a, b). While no difference in *in planta* gene induction and H3K27me3 levels could be observed between *Starship* regions and other AGRs (Fig. 3c, d), the average TE density and expression were higher in *Starship* regions than in other AGRs (Fig. 3e–g). TEs in many fungi are typically inactivated by repeat-induced point mutation (RIP) that introduces cytosine-to-thymine mutations during sexual stages[46,47]. The majority of TEs in other AGRs and core regions showed the signature of RIP (positive composite RIP index (CRI) values[48]), but the majority of TEs in *Starship* regions did not (Fig. 3h), consistent with their expression levels (Supplementary Fig. S5). *V. dahliae* AGRs have furthermore been characterized by segmental duplications[37] that physically interact in the nucleus[42]. Interestingly, consistent with the segmental duplication pattern (Figs. 2d, 3i), such bipartite long-range chromatin interactions occur between the *Ar1h1* and *Ar4h1 Starships*, between nested *Starships* within the *Ar1h1* and *Ar4h1 Starships*, and between the *Se1h1 Starship* and syntenic *Starship* regions in the JR2 genome (Fig. 3j). Collectively, our data indicate that *Starships* in *V. dahliae* are strongly associated with traits that characterize AGRs.

### *Starships* carry virulence cargo genes

To assess whether previously characterized virulence genes besides *Ave1* and *Av2* appear as cargo, we queried the *Starships* for *Verticillium* genes that are described in the Pathogen–Host Interactions Database (PHI-base)[49]. This revealed that the G-LSR2 region, proposed to confer

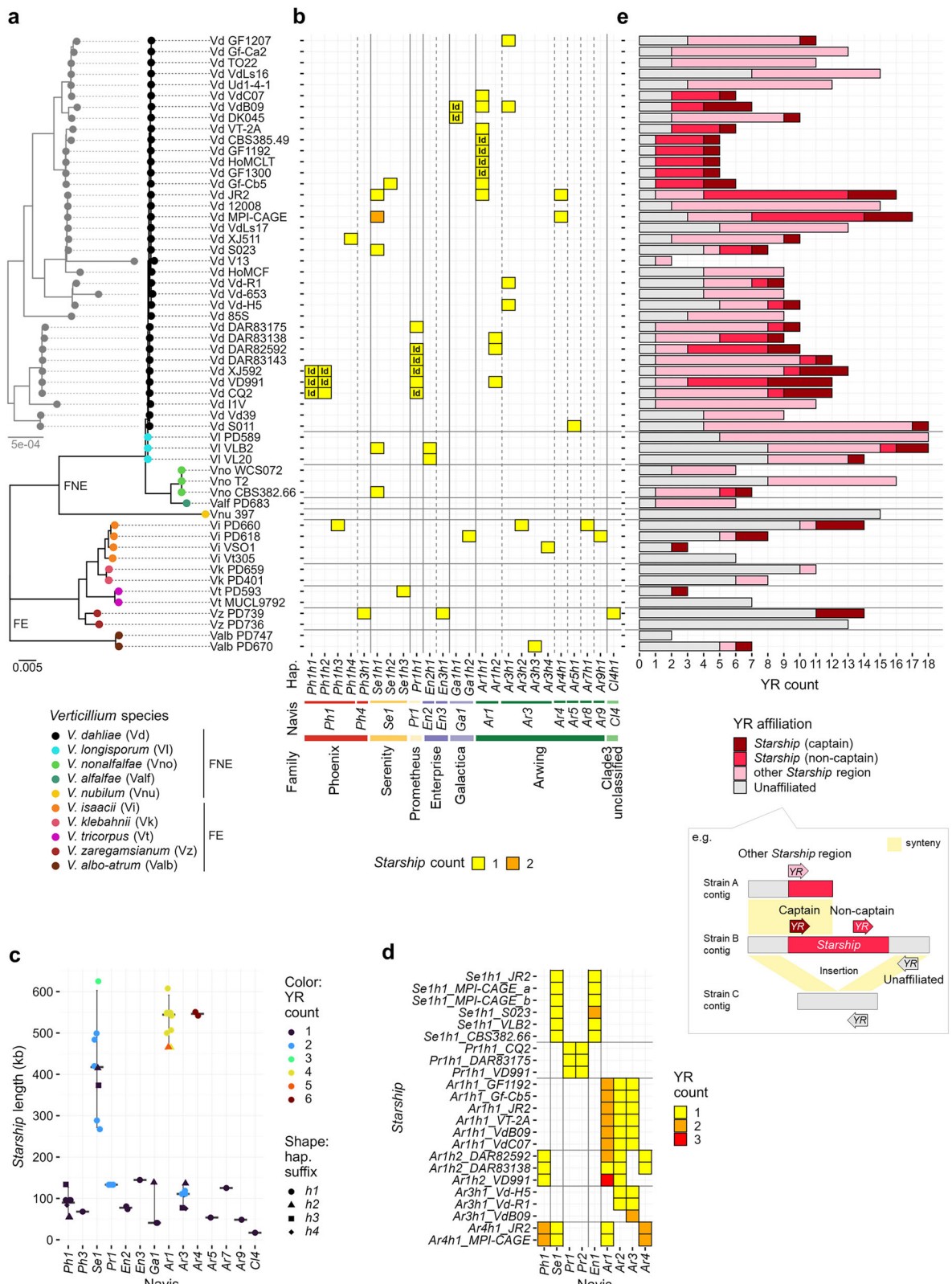

cotton-specific virulence in *V. dahliae*[50], occurs in a *Ph1h1 Starship*, while the Vl43-LS region that affects *V. longisporum* virulence[51] occurs in a *Se1h1 Starship* (Fig. 3a, b). In addition, several *Starships* (*Se1h1*, *Se1h2*, *Se1h3*, and *En3h1*) contain orthologs of other genes that were shown to contribute to virulence. These genes encode transporters, *β*-tubulin, and regulators of infection structure morphogenesis and pH-signaling[52–56] (Supplementary Fig. S6 and Supplementary Data 7),

demonstrating that *Verticillium Starships* carry diverse virulence-associated cargo genes.

### *Starship* dynamics within and between *Verticillium* genomes

HGT among *Verticillium* spp. may have contributed to the shaping of AGRs[57]. To detect possible *Starship* transfer between *Verticillium* spp., we utilized an implicit phylogenetic method that identifies segments

**Fig. 1 | Diverse *Starships* populate the *Verticillium* genus. a** The tree in black shows the phylogeny of the 56 strains used in this study across the *Verticillium* genus, divided into the Flavexudans (FE) and Flavnonexudans (FNE) clades[28], based on whole-genome sequence alignments. Circle colors indicate the ten *Verticillium* spp. while the label comprises species abbreviation followed by strain name. The tree in gray shows only the *V. dahliae* strains at increased resolution. Scale bars indicate phylogenetic distances expressed as nucleotide substitutions per site. **b** Repertoires of *Starship* haplotypes (hap.) per strain. *Starships* were classified according to previous studies[5,6]. Columns indicate *Starship* haplotypes defined by *k*-mer similarity and named according to captain navis and family (Supplementary Fig. S1b), whereas heatmap colors show haplotype member counts. "Id" refers to identical *Starships* (coverage and nucleotide sequence identity >98%) within a haplotype. **c** Size of the different *Verticillium Starships*. Points indicate individual *Starships* listed in Supplementary Data 3. Gray crossbars and error bars indicate the median and 95% confidence interval range of *Starship* lengths for each navis. **d** Captain/captain-like tyrosine recombinase (YR) navis repertoires in *Starships* with multiple YR genes. **e** Captain/captain-like YR gene classification per strain where "*Starship* (captain)" indicates YR genes located as the first gene at the 5′-terminus of a *Starship* for which both borders could reliably be identified with the Starfish tool[6], "*Starship* (non-captain)" refers to YR genes located at other sites in a *Starship*, suggesting nested *Starships* with unidentified boundaries. Furthermore, "other *Starship* region" indicates YR gene presence in regions that could not reliably be identified as *Starship* with Starfish, but that are syntenic to reference *Starships*. Finally, "unaffiliated" refers to YR genes that cannot reliably be affiliated with a *Starship* region.

with increased sequence similarity over the genome-wide average nucleotide identity (ANI)[58]. This analysis revealed that several *Starships*, such as those belonging to haplotypes *Se1h1* and *Ar1h1*, as well as syntenic *Starship* regions have higher ANI levels than the genome-wide average between the *Verticillium* species in which they occur (Fig. 4a, b and Supplementary Data 8). The most conspicuous is the *Ar1h1 Starship* that carries *Ave1*. The *Ar1h1 Starship* and its syntenic regions share over 99% identity over 98% coverage across 0.5 Mb between *V. dahliae* and *V. nonalfalfae*, while the genome-wide average identity between these species is only 93% (Fig. 4c and Supplementary Data 8).

To further support *Starship* mobility, we compared synteny of orthologous *Starship* flanking regions, revealing that *Se1h1* and *Ar1h1 Starships* occur in several non-homologous genomic regions in different *Verticillium* clades (Fig. 4c, d and Supplementary Fig. S7). For instance, the *Ar1h1 Starship* and its syntenic regions that occur in nine *V. dahliae* strains and in two *V. nonalfalfae* strains collectively inserted into regions that are orthologous to five different JR2 chromosomes (Fig. 4d), suggesting at least four *Ar1h1 Starship* movements. *Se1h1 Starships* in three *V. dahliae* strains, one *V. longisporum* strain, and one *V. nonalfalfae* strain collectively occur in regions that are orthologous to four different JR2 chromosomes (Supplementary Fig. S7c). *Se1h1 Starships* in some strains appear to have been generated by rearrangements between regions orthologous to two JR2 chromosomes (chr2 and chr5) but have been inserted into two other different chromosomal regions (JR2 chr1 and chr4), suggesting at least two movements (Supplementary Fig. S7c).

Comparisons among orthologous *Starship* regions across *Verticillium* strains revealed various diversification patterns (Supplementary Fig. S8). Besides nested *Starship* insertions (e.g., *Ar3h1 Starship* in *Ar1h1 Starship*, Supplementary Fig. S8a and Fig. 1c, d), we typically observed gain and loss of cargo elements in TE-enriched *Starships* (e.g., *Pr1h1 Starships*, Supplementary Fig. S8b, c). We furthermore observed invasion of orthologous sites by different *Starships* that contain orthologous captains, but otherwise lack synteny (e.g., *Ar1h1* and *Ar1h2 Starships*, Supplementary Fig. S8d), which points towards independent insertions as orthologous captains target particular sequences as *Starship* insertion sites[6,8,59].

### Cross-order horizontal *Starship* transfer

To detect possible horizontal *Starship* transfer between *Verticillium* and other fungi, we queried all Pezizomycotina genomes available in the GenBank WGS database (10,113 genomes from 668 genera, Supplementary Fig. S9a and Supplementary Data 9) with *Verticillium Starships*. Intriguingly, regions syntenic to *Ph1h1* and *Ph1h2 Starships* that were detected in only three *V. dahliae* strains (CQ2, VD991, and XJ592) (Fig. 4a, b), were found in the genomes of various *Fusarium* species, some of which contain regions syntenic to the G-LSR2 region (*Ph1h1 Starship* cargo) that was associated with *V. dahliae* virulence[50] (Fig. 5a and Supplementary Fig. S9b). Since *Fusarium* belongs to Hypocreales and synteny to these *Verticillium Starships* lacks in non-*Verticillium* genomes of the Glomerellales to which *Verticillium* spp.

belong, these results suggest horizontal transfer of *Ph1h1* and *Ph1h2 Starships* between *Verticillium* and *Fusarium*.

To identify the directionality of horizontal transfer, we analyzed a *k*-mer-based phylogeny of *Starships*[60]. To this end, we identified 52 *Starships* in 283 high-quality *Fusarium* genome assemblies (Supplementary Data 10, 11) and analyzed their phylogeny together with the 54 *Verticillium Starships*, revealing that most *Starships* clustered according to genus (Fig. 5b). In contrast, the *Verticillium Ph1h1* and *Ph1h2 Starships* phylogenetically localized in the *Fusarium Starship* clade (Fig. 5b), suggesting *Fusarium* to *Verticillium* transfer. Moreover, we identified a near-identical *Ph1h2 Starship* that shares over 99% nucleotide identity over 50 kb in three *V. dahliae* strains and in *F. keratoplasticum* (Fig. 5c). The *Starship* flanking regions are syntenic in three phylogenetically clustered *V. dahliae* strains, suggesting that the *Ph1h2 Starship* invaded *V. dahliae* from *F. keratoplasticum* before these strains diverged (Fig. 5c).

To further explore potential *Starship*-associated HGTs overlooked by the synteny search, we tested HGT signatures for each *Starship* cargo element using the Alien index (AI) that detects higher similarity to outgroup than ingroup orthologs[61]. We queried orthologs of *Verticillium* genes in the Pezizomycotina genomes and calculated AI scores with non-*Verticillium* Glomerellales genera as ingroup and the other 74 orders as outgroup. This showed that the median AI for JR2 genes is higher in *Starship* regions (45) than in other AGR (0) and core regions (−59), and that the proportion of genes with AI ≥45 (indicative of HGT) is higher in *Starship* regions (19%) and other AGRs (17%) than in core regions (4%), suggesting cross-order HGT in *Starship* regions and other AGRs (Fig. 5d). Among the virulence-associated cargo genes, *Av2* as well as the G-LSR2 genes showed AI >45 with hits in the various *Fusarium* spp. of the Hypocreales while lacking ingroup hits (Fig. 5e). Horizontal *Av2* transfer was further supported by phylogenetic analyses in which *Verticillium* orthologs were detected in a clade within a larger clade of *Fusarium* orthologs (Supplementary Fig. S10a, b). *Fusarium Av2* orthologs were found proximal to captain-like YR genes, which suggests *Starship* association (Supplementary Fig. S10c). We also tested the HGT signature of transcriptionally active *Verticillium* TEs[62] that occur in *Starship* regions (Supplementary Fig. S11a). Surprisingly, seven out of nine cargo TEs showed AI >45 with the best outgroup hits in five orders, including Hypocreales, while six of them lacked ingroup hits (Fig. 5e). Orthologs of the five cargo TEs also occurred in *Fusarium Starships* (Supplementary Fig. S11b), suggesting horizontal transfer via *Starships*. Collectively, our results provide evidence for horizontal transfer of various *Starship* cargo elements between *Verticillium* and fungi that even belong to other orders.

### *Starships* mediate de novo gene birth

*Starships* carry diverse lineage-specific genes[7], yet their origin remains enigmatic. We pursued the evolution of the *NLP6* effector gene that occurs in a limited number of *V. dahliae* strains, including VdLs17[63], and was detected near a *Starship* region (Supplementary Fig. S12a). NLP6 belongs to the family of necrosis- and ethylene-inducing peptide

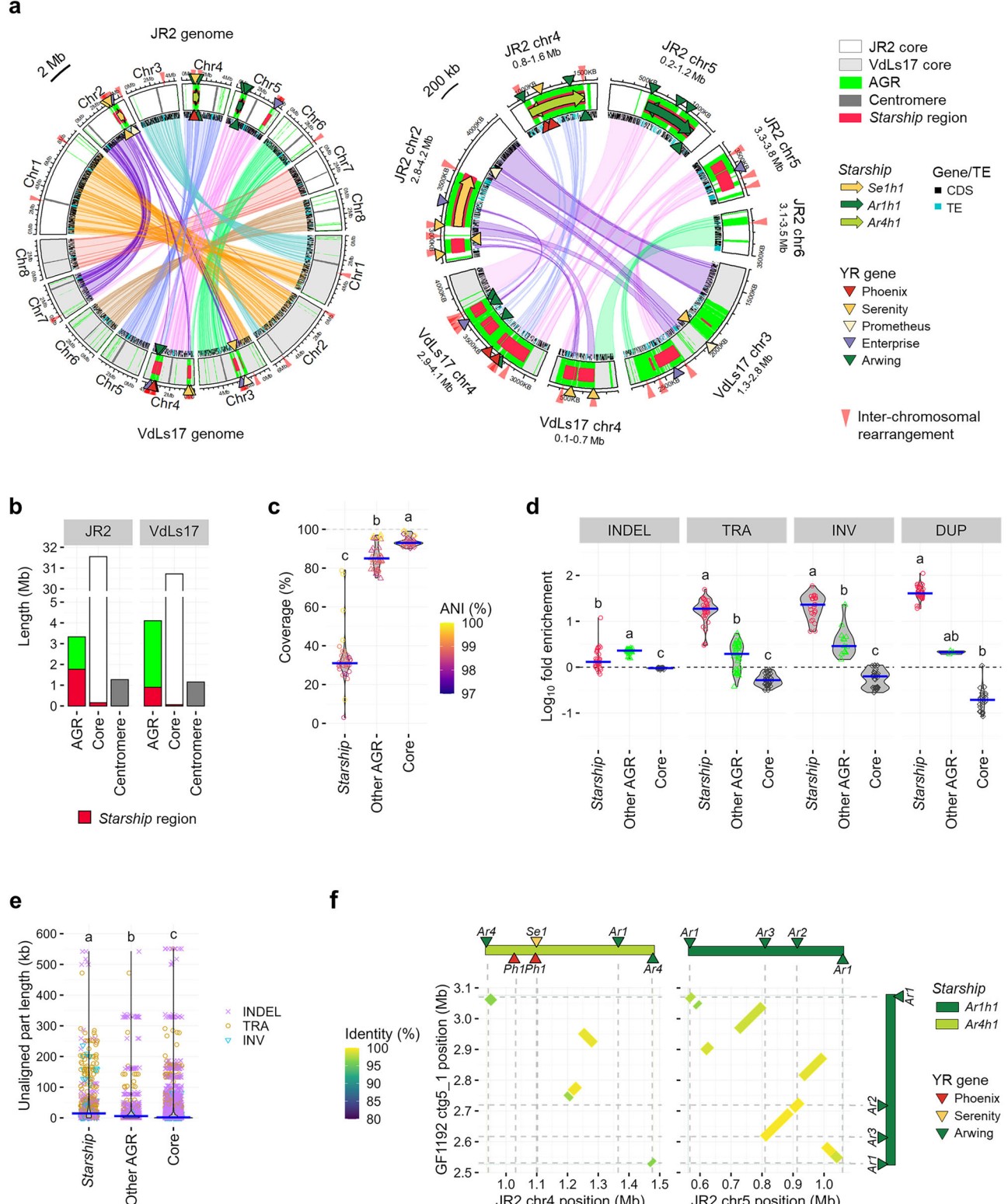

1 (Nep1)-like proteins (NLPs), many members of which confer virulence[63,64]. As multiple NLP paralogs occur in *Verticillium* genomes[63], we first analyzed the phylogeny of NLP homologs in the 56 *Verticillium* genomes. NLP6 occurs in the same clade as NLP3 orthologs which were detected in all 56 strains (Fig. 6a). Intriguingly, NLP6 is more closely related to NLP3 orthologs of *V. nubilum* and species of the FE clade than to those of *V. dahliae* or other species of the FNE clade (Fig. 6a). The similarity between NLP3 orthologs and NLP6 occurs at the C- but not at the N-terminus (Supplementary Fig. S12b). Rather, the

3' end (115-450 nt) of *NLP6* shares 90% nucleotide identity with the 3' end (379-714 nt) of *V. nubilum NLP3*, which greatly exceeds the genome average (82%) (Fig. 6b, c, Supplementary Fig. S12c, and Supplementary Data 8), suggesting that the 3' part of *V. dahliae NLP6* is derived from a horizontally transferred *NLP3* ortholog. To explore the origin of the 5' part (1-114 nt) of *NLP6*, we queried the *Verticillium* genomes and detected hits with 96% nucleotide identity in non-coding regions in *Se1h1 Starships* in four *Verticillium* species, which we termed *tNLP6* (for *truncated NLP6*) (Fig. 6d–f). Intriguingly, three *tNLP6* copies were

**Fig. 2 | *Starships* are hotspots of genomic rearrangements. a** Circular plots showing the locations of *Starship* regions, adaptive genomic regions (AGRs), and genomic rearrangements between *V. dahliae* strains JR2 (upper eight chromosomes) and VdLs17 (lower eight chromosomes). Tracks are filled with colors representing either core, AGR, or centromeric regions, overlaid with bold lines and arrows representing *Starship* regions and *Starships* colored by haplotype. Overlapping arrows in a single region indicate that the *Starship* orientation is not determined due to the presence of captains at the 5' end of both strands. Regular triangles point to the captain and captain-like tyrosine recombinase (YR) genes, colored by family and annotated with navis identification (ID). The overlaps among these elements and rearrangement breakpoints are shown in Supplementary Fig. S2 at a higher resolution. Colored bands at the inner edge of tracks represent genetic elements grouped into protein coding sequence (CDS) and transposable element (TE). Ribbons connect syntenic regions (>80% nucleotide sequence identity over 10 kb). **b** Total length of genomic compartments in *V. dahliae* strains JR2 and VdLs17. **c** Violin plots depicting the sequence alignment coverage determined by comparing JR2 genomic compartments against 35 *V. dahliae* genomes (Supplementary Fig. S3). Points indicate the coverage, and the color represents genome-wide average nucleotide identity (ANI) for each genome alignment ($n = 35$), while blue crossbars indicate the median values. Different letter labels indicate significant differences (two-sided Dunn's test, adjusted $p < 0.05$). **d** Violin plots depicting the fold-enrichment of structural variations (SVs) (insertion and deletion (INDEL) without size definition, translocation (TRA), inversion (INV), and duplication (DUP)) in the three genomic compartments compared with the genome-wide average. SVs were determined by comparing the genome of *V. dahliae* strain JR2 against the 35 *V. dahliae* genomes. Points indicate the fold-enrichment determined for each genome ($n = 35$), while blue crossbars indicate the median values. Different letter labels indicate significant differences for each genetic variation (two-sided Dunn's test, adjusted $p < 0.05$). **e** Violin plots depicting length of unaligned regions accompanied by SVs in JR2 genomic compartments. Points indicate the length of every gap ($n = 644$ in *Starship* regions, $n = 1163$ in other AGRs, and $n = 8877$ in core regions) found between the JR2 genome and 35 *V. dahliae* genomes in a color and shape representing SV type, while blue crossbars indicate the median values. Different letter labels indicate significant differences (two-sided Dunn's test, adjusted $p < 0.05$). **f** *Starship* rearrangements between the genomes of *V. dahliae* strains JR2 (X-axis) and GF1192 (Y-axis). Diagonal lines indicate synteny, while the color represents nucleotide sequence identity. Bars and triangles aligned to the plots indicate the positions of *Starships* and YR genes. The left and right plots are superimposed in Supplementary Fig. S4b to make the breakpoints of inter-chromosomal rearrangements clearer.

detected around *NLP6* in VdLs17 (Fig. 6e). Collectively, these results suggest that *NLP6* could be formed through the fusion of a non-coding *Starship* element with the 3' part of an *NLP3* ortholog through a genomic rearrangement (Fig. 6g). Interestingly, although a role of *NLP3* in fungal virulence could not be demonstrated, deletion of *NLP6* results in reduced symptom development in *V. dahliae*-inoculated tomato plants (Fig. 6h and Supplementary Fig. S13). Collectively, these results demonstrate that *Starships* mediated the emergence of a virulence-associated lineage-specific effector gene in *V. dahliae*.

## Discussion

Here, we reveal the unprecedented impact of *Starships* on fungal genome evolution through four main findings. First, every *Verticillium* genome contains *Starships* or their remnants. Secondly, these *Starships* compose the vast majority of AGRs that govern pathogenicity on plant hosts. Thirdly, extensive horizontal *Starship* transfer occurred between *V. dahliae* and phylogenetically diverse fungi. Fourthly, *Starships* contributed to the de novo generation of a novel virulence gene. Thus, *Starships* are not only instrumental for *Verticillium* genome evolution, but also fundamental to evolutionary innovations that led to plant pathogenicity.

Thus far, little is known about *Starship* abundance in individual fungal genomes. We identified 24 *Starship* haplotypes in 56 *Verticillium* genomes. These *Starships* generally occur in multiple *Verticillium* strains, indicating invasion before strain diversification. While some remained intact, others got disrupted by genomic rearrangements. The abundance of *Starships* exceeds the abundance previously recorded in *A. fumigatus*, where 20 *Starships* and their remnants were identified in a survey of 519 strains, and not all strains contained *Starships*[9]. In *M. phaseolina*, four *Starships* and remnants were identified in 12 isolates[7]. Overall, our study reveals an unprecedented abundance of *Starships* across *Verticillium* genomes.

We could unequivocally demonstrate that *Starships* make up about half of AGRs in the *V. dahliae* JR2 genome, while the remaining AGRs could not be assigned to *Starships*. AGRs were initially identified as lineage-specific genomic regions by comparisons among *V. dahliae* strains[32], while additional AGRs were subsequently identified based on their characteristic chromatin profile[30]. Interestingly, the *Starship* regions identified in this study correspond to the initially identified AGRs. However, besides their chromatin profile, like *Starships*, also the non-*Starship* AGRs are enriched in horizontally transferred genes and genomic rearrangements. Thus, we speculate that these AGRs may be derived from unidentified *Starships* that became disrupted by genomic rearrangements. In support of this hypothesis, *Verticillium* genomes

likely still contain many unidentified *Starship* regions, given 211 captain-like YR genes unassigned to known *Starship* regions in the 56 genomes, the majority (127/211) of which occur in AGRs (Supplementary Data 4). Overall, we conclude that *Starships* are the major constituents of AGRs in *V. dahliae*.

Many filamentous plant pathogens carry plastic genomic regions similar to AGRs[20–24], but how these regions evolved remains elusive[26]. Although several *Starships* have been identified in Pezizomycotina plant pathogens[7,12,14,15,65–68], the association of these *Starships* with the typical plastic genomic regions has not been noted previously. The evolution of plastic genomic regions is known to be associated with the activity of smaller TEs[20,26]. Our results show that TEs in *Starship* regions are more highly expressed and show weaker RIP signatures than those in the other regions of the *V. dahliae* genome. RIP mutates TEs during sexual stages in Pezizomycotina fungi[47]. As *V. dahliae* is presumed to reproduce asexually[69], it has been hypothesized that TEs showing RIP signatures could have been mutated in its sexually active ancestor in which RIP was active[30]. We have previously shown that TEs in lineage-specific AGRs, which correspond to *Starship* regions identified in this study, are enriched in evolutionarily younger TEs that proliferated after the speciation of *V. dahliae*, whereas other regions are enriched in older TEs that proliferated before speciation[37]. Therefore, the enrichment of transcriptionally active TEs in *Starship* regions could be due to their invasion of *V. dahliae* after the inactivation of TEs in other genomic regions. In support of this hypothesis, our results suggest the horizontal transfer of active TEs in *Starships*. These findings highlight the close relationships among plastic genomic regions, *Starships*, and the activity of other TEs.

HGT has driven virulence evolution in diverse fungal pathogens[70,71]. HGT between phylogenetically close fungi is thought to be mediated by anastomosis and horizontal chromosome transfer, while the mechanisms of HGT between phylogenetically distant fungi remain elusive[70,71]. Our phylogenetic analyses suggest the horizontal *Starship* transfer between *V. dahliae* and *V. nonalfalfae* on the one hand, and distantly related *Fusarium* fungi on the other hand. In addition, *Starship* regions in *V. dahliae* are enriched in cargo elements horizontally transferred to/from fungi belonging to further diverse Pezizomycotina orders. Thus, while previous studies have shown *Starship*-mediated HGT between fungi within the same order[8,10–16], our study extends this observation by revealing that *Starships* mediated HGT between phylogenetically distant fungi of different orders. There are many reports of cross-order HGT between diverse Pezizomycotina pathogens, though the mechanism remains generally unknown[72–78]. Further identification of *Starships* in diverse taxa may

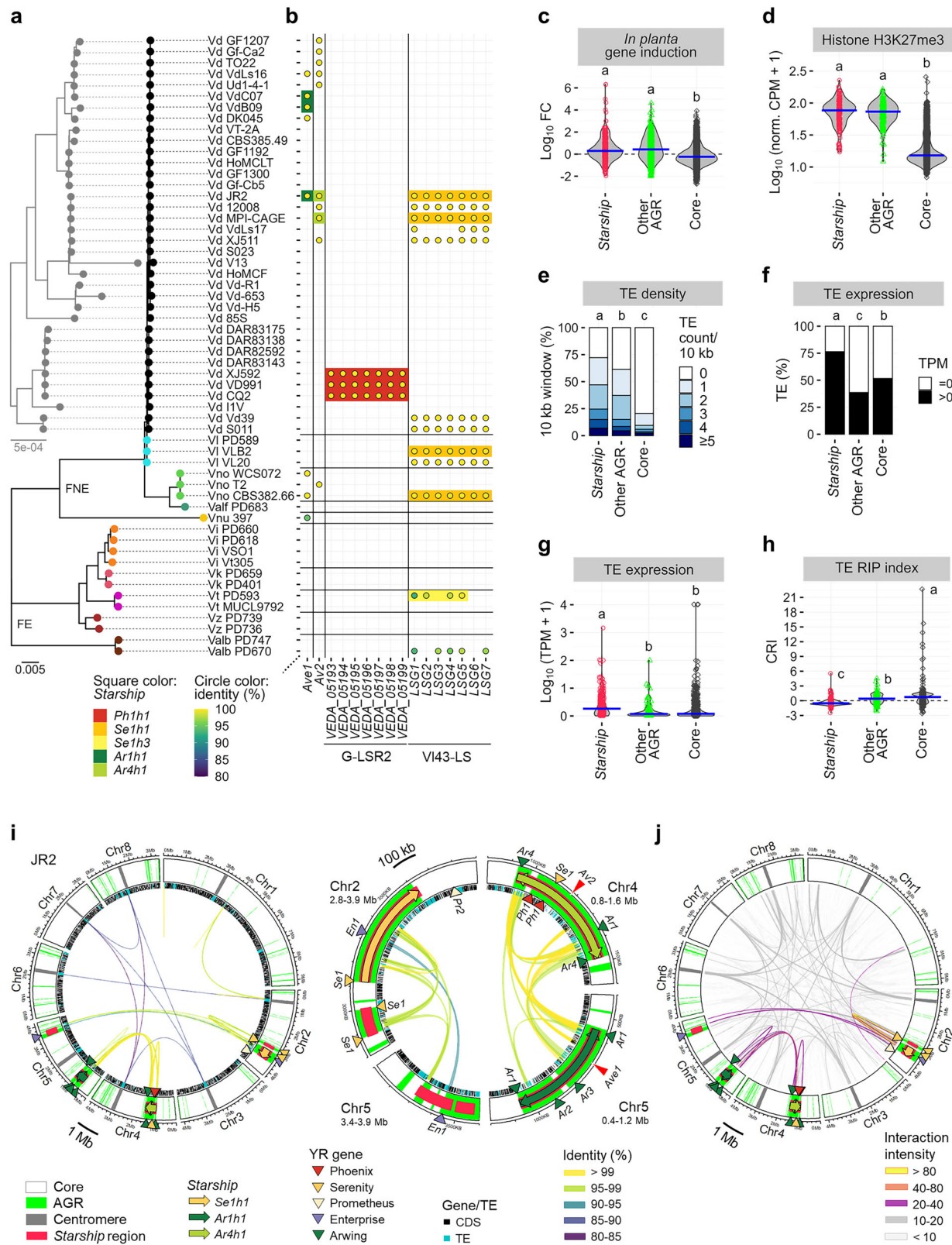

help to understand the extent to which *Starships* have mediated such HGT events.

Pathogens exploit diverse effectors to colonize hosts, but it often remains unclear where, when, and how such effectors evolved, given that they are often considered lineage-specific inventions[79]. Since pathogen effectors often lack conserved protein domains, it has been speculated that such effectors may have evolved de novo[79]. De novo

genes are defined to originate, at least in part, from non-coding DNA sequences, but the underlying evolutionary mechanisms remain poorly understood[80,81]. Our results suggest that a *Starship* contributed to the emergence of a virulence effector gene from a non-coding cargo sequence by the fusion with a section of a conserved effector gene. Given the strong association of *Starships* with plastic genomic regions enriched in (a)virulence effector genes, further elucidation of

**Fig. 3 | *Starships* show typical traits of the plastic genomic regions. a** *Verticillium* phylogeny as described in Fig. 1a. **b** Presence or absence of (a)virulence-associated genes in the 56 *Verticillium* strains. Square colors indicate *Starships* in which the genes were detected, while the other orthologs found in syntenic *Starship* regions are not colored. Circles indicate the presence of orthologs with 100% gene coverage and the fill color representing nucleotide identity. **c** Violin plots depicting the *in planta* expression induction of genes residing in the three genomic compartments in *V. dahliae* strain JR2. Induction levels are represented by fold-changes (FC) of gene expression *in planta* (*Arabidopsis thaliana*) versus in vitro (potato dextrose broth, PDB). Points indicate FC values for each gene ($n = 450$ in *Starship* regions, $n = 547$ in other adaptive genomic regions (AGRs), and $n = 10,638$ in core regions), while blue crossbars indicate median values. Different letter labels indicate significant differences (two-sided Dunn's test, adjusted $p < 0.05$). **d** Violin plots depicting histone H3K27me3 levels for the three JR2 genomic compartments expressed as ChIP-Seq read counts per million (CPM) normalized by bin length. Each point indicates the value for a bin (-10 kb) ($n = 192$ in *Starship* regions, $n = 160$ in other AGRs, and $n = 3145$ in core regions), while blue crossbars indicate median values. Different letter labels indicate significant differences (two-sided Dunn's test, adjusted $p < 0.05$). **e** Stacked bar plots depicting the transposable element (TE) density in the JR2 genomic compartments over 10 kb windows ($n = 188$ in *Starship* regions, $n = 153$ in other AGRs, and $n = 3125$ in core regions). Different letter labels indicate significant differences (two-sided Dunn's test, adjusted $p < 0.05$). **f** Proportion of expressed (transcripts per million (TPM) >0) and non-expressed (TPM = 0) TEs in the JR2 genomic compartments ($n = 284$ in *Starship* region, $n = 202$ in other AGRs, and $n = 1261$ in core regions). Different letter labels indicate significant differences (two-sided Fischer's test with Bonferroni correction, adjusted $p < 0.05$). **g** Violin plots depicting the TE expression levels in the three JR2 genomic compartments for *V. dahliae* cultivated in PDB. Points indicate TPM values for individual TEs excluding non-expressed ones ($n = 217$ in *Starship* regions, $n = 78$ in other AGRs, and $n = 650$ in core regions), while blue crossbars indicate median values. Different letter labels indicate significant differences (two-sided Dunn's test, adjusted $p < 0.05$). **h** Violin plots depicting repeat-induced point mutation (RIP) signature of TEs in the JR2 genomic compartments. Points indicate composite RIP index (CRI)[48] for individual TEs ($n = 284$ in *Starship* regions, $n = 202$ in other AGRs, and $n = 1261$ in core regions), while blue crossbars indicate median values. Different letter labels indicate significant differences (two-sided Dunn's test, adjusted $p < 0.05$). **i** Locations of *Starships* and AGRs in the JR2 genome. See Fig. 2a legend for the details of symbols. Ribbons connect pairs of segmentally duplicated regions that share >80% nucleotide sequence identity over 10 kb, with a color representing identity. **j** Long-range chromatin interactions in the JR2 genome. Ribbons connect genomic regions that are separated in the genome but physically interact in nuclei by colors that represent the intensity.

*Starship*-mediated gene evolution can be fundamental to understanding how novel virulence genes evolve.

## Methods

### Genome assemblies and annotations

Genome sequences were collected from public databases or generated in this study (Supplementary Data 1, 9, 10). Fungal genomic DNA was extracted and sequenced using Oxford Nanopore Technologies as previously described[34]. Sequencing reads were assembled with Canu version 2.2[82], and genome assembly quality was evaluated with BUSCO version 5.7.0 with the datasets eukaryota_odb10 and glomerellales_odb10[83,84]. The genome-wide average nucleotide identity (ANI) was calculated with FastANI version 1.33[85] that maps query genomic sequences fragmented to 3 kb to a reference genome and calculates the average of the maximum identity of the mapped regions.

Genes and TEs in *V. dahliae* strain JR2 were identified previously[38,44]. In all other genomes, repetitive sequences were predicted with RepeatModeler version 2.0.5[86] and were soft-masked with RepeatMasker version 4.1.5[87]. Genes were then predicted with the BRAKER version 3.0.8 pipeline C[88], which trains GeneMark-EP+ version 4.72_lic[89] and AUGUSTUS version 3.0.8[90] with information on splice sites, start, stop, and coding features using reference fungal proteins in OrthoDB version 11[91]. Orthologous gene groups were identified by eggNOG-mapper version 2.1.12[92] using eggNOG database version 5.0.2[93]. TEs were further characterized with the TE annotation pipeline EDTA version 2.2.1[94] with curated TEs of *V. dahliae* VdLs17[62].

### Identification of *Starships* and *Starship* regions

*Starships* and captain/captain-like tyrosine recombinase (YR) genes were identified with Starfish version 1.0.0 according to the standard workflow[6]. First, the captain candidate YR genes were identified de novo from the genomes using the Starfish gene finder module with MetaEuk version 6.a5d39d9[95] and HMMER version 3.3.2[96] based on sequence similarity to the Pezizomycotina captain/captain-like YR database attached to Starfish. The identified YR genes were grouped into families as defined previously[6] based on a similarity search with HMMER version 3.3.2[96]. The YR genes were then grouped into naves by clustering of *Verticillium* YRs using MMseqs2 version 14.7e284 easy-cluster[97] with thresholds of 50% amino acid sequence identity and 25% coverage. Each *Verticillium* YR navis was named with two letters of the YR family followed by an identifier number. Then, *Starship* candidates were identified via multiple sequence alignments among the 56

*Verticillium* genomes or among the 283 *Fusarium* genomes that were selected based on a relatively low number of contig/scaffold numbers, suggesting a relatively highly contiguous genome assembly (Supplementary Data 2, 10), by the Starfish element finder module using BLAST version 2.12.0+[98] and MUMmer version 4.0.0rc1[99]. After the initial screening by Starfish, confident *Starships* were curated by (1) removing *Starship* candidates that lack flanking synteny over 20 kb against orthologous sites without *Starships* through manual inspection with Starfish pairViz, (2) removing *Starship* candidates that contain serial N stretches ($N \geq 3$) in scaffolds, and (3) unifying redundantly detected *Starships* due to the presence of YR genes on both strands around the 5' ends. The *Starships* were grouped into haplotypes based on the captain navis and nucleotide *k*-mer similarities detected by sourmash version 4.8.3[100] with Starfish sim with default *k*-mer size 510[6,9], followed by the clustering using mcl version 14-137[101] with minimal similarity threshold 0.05. Each *Verticillium Starship* haplotype was named after the captain navis ID followed by a suffix. Genomic regions orthologous to *Starship* insertion sites across *Verticillium* strains (Supplementary Fig. S7) were identified by the Starfish region finder module based on the presence of low copy number (up to 5) eggNOG orthologs.

*Starship* regions were annotated by two approaches. First, regions downstream of YRs were annotated as *Starship* regions using Starfish extend[6] with default settings using BLAST version 2.12.0+[98] based on similarity against curated *Starships* (Supplementary Data 12). Second, other *Starship* regions were identified irrespective of the presence of YR genes based on synteny against curated *Starships* through the alignments with nucmer of MUMmer version 4.0.0rc1 with options maxmatch and minimal alignment length 10,000, followed by filtering using delta-filter with the threshold of >80% nucleotide identity[99] (Supplementary Data 13). *Starships* identified in this study (Supplementary Data 3, 11) were used as reference. The *Starship* regions identified by the two approaches were merged with bedtools version 2.30.0[102] to determine the final coordinates of *Starship* regions (Supplementary Data 5). The pairwise sequence alignments between each *Starship* and syntenic *Starship* regions in each genome (Fig. 4b) were also performed using nucmer with the same settings.

The genomic compartments of *V. dahliae* strains JR2 and VdLs17 were assigned based on the overlap of *Starships* and *Starship* regions with the previously assigned AGRs, core genomic regions[30], and centromeres[103]. AGRs in the JR2 genome were identified by chromatin profiling[30], while AGRs in the other strains refer to regions syntenic to AGRs in JR2 and regions absent in JR2[42]. To identify AGRs in strains

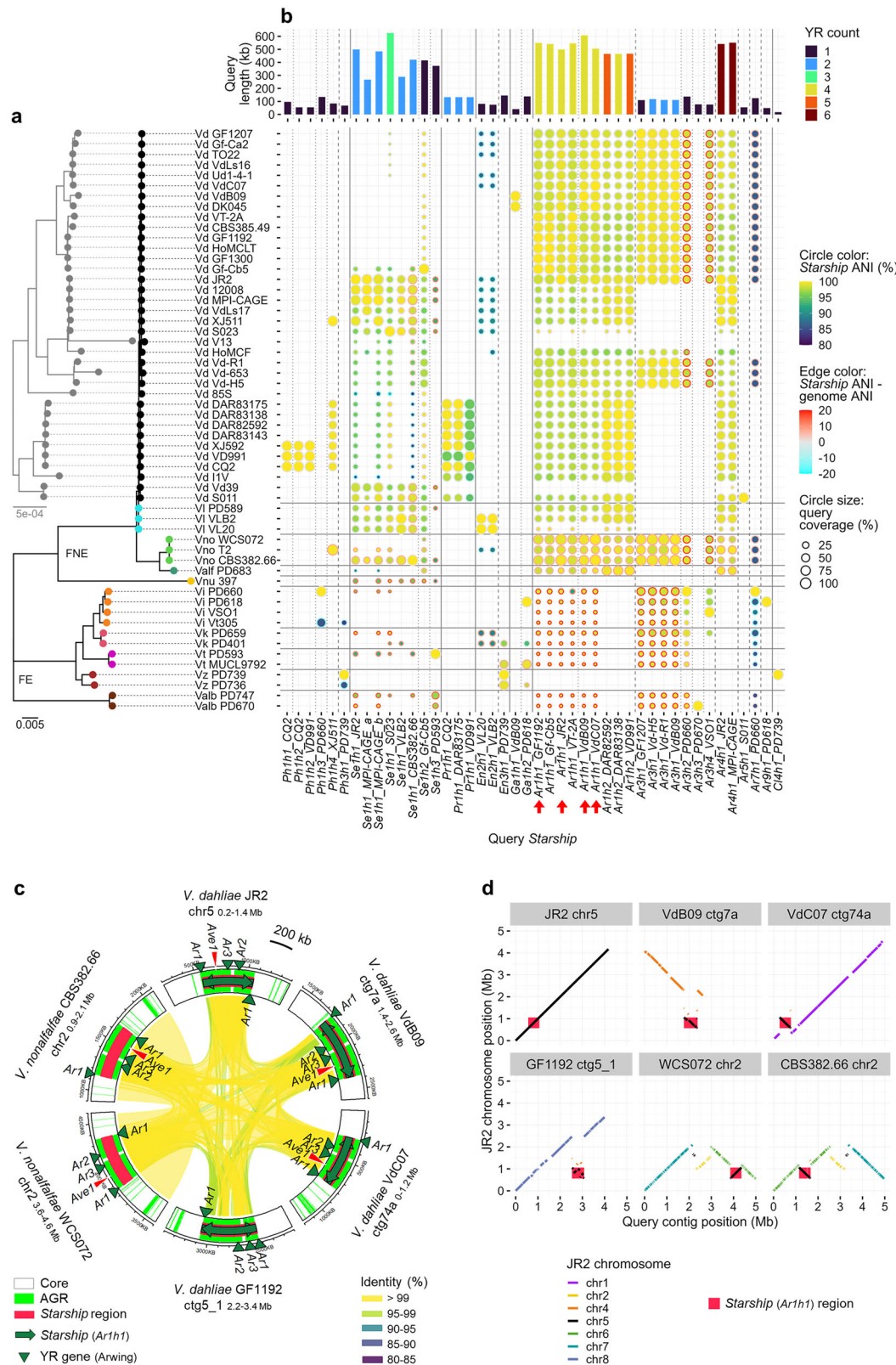

**Fig. 4 | *Starship* mobility among *Verticillium* genomes. a** *Verticillium* phylogeny as described in Fig. 1a. **b** Occurrence of diverse *Starships* across the *Verticillium* genus. Columns indicate different *Starships* with their length and number of captain and captain-like tyrosine recombinase (YR) genes. The heatmap indicates the query coverage, Average Nucleotide Identity (ANI) of *Starship* regions (*Starship* ANI), and the difference between *Starship* ANI and genome-wide ANI for each pairwise strain comparison to detect potential horizontal transfers. Red arrows indicate the *Starship* highlighted in (**c**). **c** Circular plot depicting the synteny among

regions of four *V. dahliae* and two *V. nonalfalfae* strains harboring orthologous *Ar1h1 Starships*. See Fig. 2a legend for the details of symbols. **d** Synteny plots between the *V. dahliae* JR2 genome and the *Ar1h1 Starship* regions in (**c**). Each plot indicates the synteny between the JR2 chromosomes (Y-axis) and each chromosome (chr) or contig (ctg) containing the *Ar1h1 Starship* (X-axis), with diagonal lines colored by syntenic JR2 chromosome. Red background indicates the coordinates of *Ar1h1 Starship*.

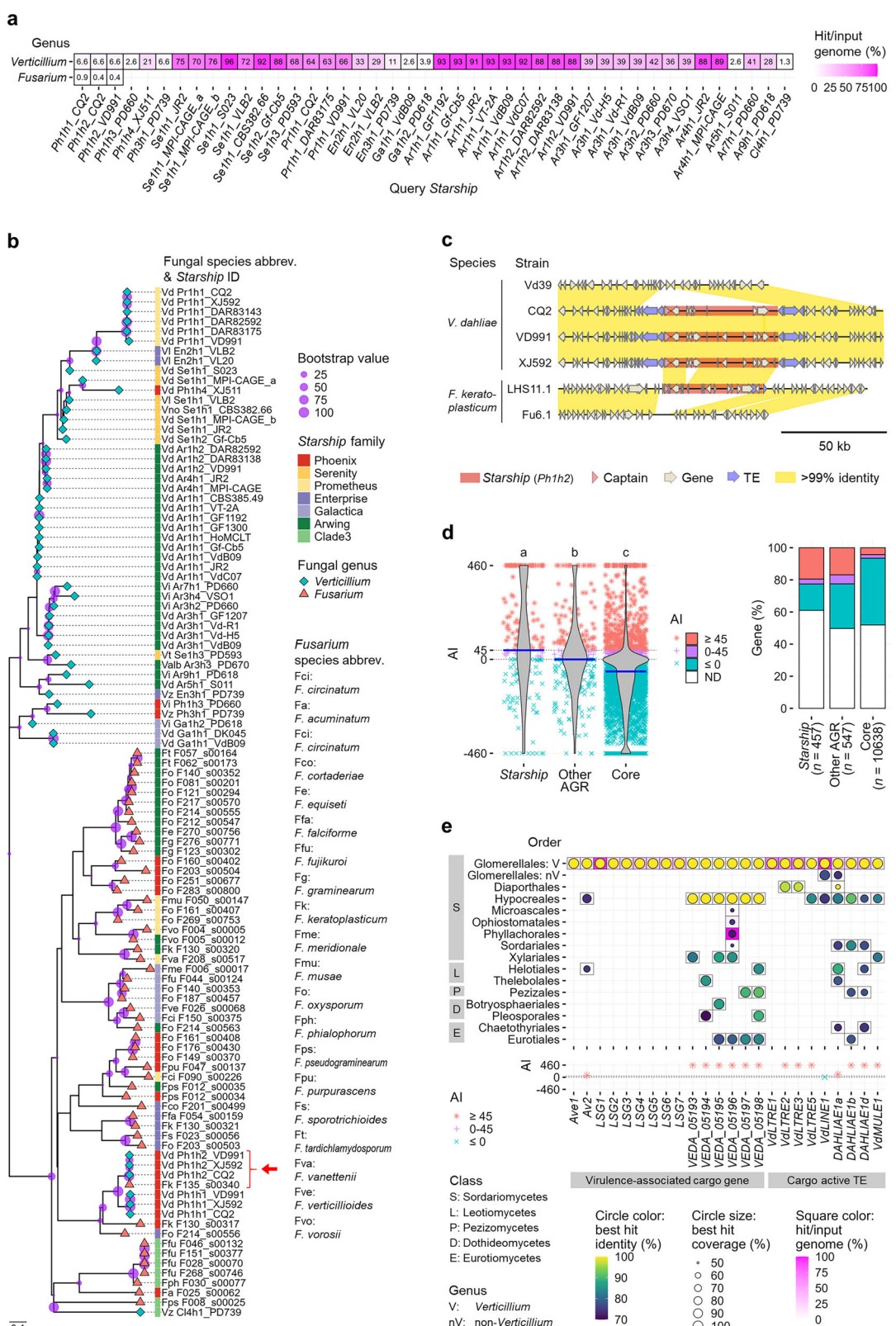

other than JR2, genome alignments were performed with nucmer of MUMmer version 4.0.0rc1[99], followed by filtering of alignments with >80% nucleotide identity over 1 kb. The regions aligned to the JR2 AGRs and not aligned to the JR2 genome were merged with bedtools version 2.30.0[102] and then filtered by the length threshold of >1 kb to determine the final AGRs. The "*Starship*" compartment refers to *Starships* and additional *Starship* regions, "other AGR" refers to AGRs that

do not belong to *Starship* regions, "centromere" refers to previously identified centromeres, and "core" refers to core genomic regions that do not belong to *Starship* regions or centromeres. Genes and TEs in the JR2 genome were assigned to genomic compartments based on the location of their midpoints. TE counts over 10 kb windows in each genomic compartment were calculated using bedtools version 2.30.0 intersect[102]. Compartments smaller than 10 kb were omitted from the

**Fig. 5 | *Starship* dynamics among fungal orders. a** Occurrence of *Verticillium Starships* across 10,113 Pezizomycotina genomes. Cell colors and labels denote the percentage of genomes with hits (e-value < 0.05 and total query coverage >50% or >30 kb) in each genus. **b** Phylogeny of *Starships* that occur in the *Verticillium* and *Fusarium* genera based on *k*-mer similarity. Scale bar indicates the Mash distance that represents the *k*-mer difference[140]. Circles at the nodes represent bootstrap values for 1000 iterations. Red arrows indicate the *Starship* highlighted in (**c**). **c** Similarity of *Ph1h2 Starships* and surrounding regions between *V. dahliae* and *F. keratoplasticum*. **d** Possibility of horizontal gene transfer (HGT) between *Verticillium* and fungi belonging to the other orders of Pezizomycotina for genes residing in the three genomic compartments in *V. dahliae* strain JR2. High Alien index (AI) values (≥45) suggest HGT, while moderate values (>0 and <45) suggest a

weak HGT possibility. Points in the left violin plots indicate AI values of individual genes (*n* = 457 (including captain and captain-like genes predicted de novo) in *Starship* regions, *n* = 547 in other adaptive genomic regions (AGRs), and *n* = 10638 in core regions), while blue crossbars indicate median values. Different letter labels indicate significant differences (two-sided Dunn's test, adjusted *p* < 0.05). The right bar plots depict the percentage of genes with high, moderate, and low AI values. ND indicates that AI was not determined because of lacking hits in both ingroup and outgroup. **e** Occurrence of homologs of *Verticillium Starship* cargo genes and transposable elements (TEs) across 10,113 Pezizomycotina genomes. Square colors represent the percentage of genomes with hits (e-value < 0.05, over 70% nucleotide identity over 50% coverage) in each group. Circles indicate the coverage and identity of the best hits. The lower panel indicates AI values as described in (**d**).

TE density analysis. The composite RIP index (CRI) of TEs was calculated as previously described[48]. Briefly, CRI was determined by subtracting the RIP substrate index ((CpA + TpG)/(ApC + GpT)) from the RIP product index (TpA/ApT), as CpA dinucleotides are preferentially targeted by RIP.

### Analysis of genomic syntenies and variations

Genomic syntenies and variations were identified with MUMmer version 4.0.0rc1[99]. Specifically, segmental duplications in the JR2 genome were identified by genome self-alignment using nucmer with options maxmatch, nosimplify, and minimal alignment length 10,000, followed by filtering with delta-filter. Inter-genomic syntenies and variations were identified by pairwise genome alignments with nucmer with option minimal alignment length 10,000, followed by filtering and analysis by dnadiff. The genome-wide alignment coverage was calculated by integrating all fragmented hits. Among the structural variations (SVs) shown in this study, insertion/deletion (INDEL) corresponds to "GAP" and "BRK" in MUMmer4, while translocation (TRA) corresponds to "JMP" and "SEQ". SV frequency was calculated by dividing the SV counts by the aligned length for each genomic compartment or genome. SV fold-enrichment was calculated by dividing the frequency in each genomic compartment by the genome-wide frequency.

### Transcriptome, epigenome, and 3D genome analyses

Global gene expression in *V. dahliae* strain JR2 was analyzed using the previous RNA sequencing (RNA-Seq) read data[104] obtained from the NCBI short read archive (SRA) under the accession numbers listed in Supplementary Data 14. RNA-Seq reads were filtered with fastp version 0.19.5[105] and mapped to the unmasked JR2 genome with STAR version 2.7.10a[106] with previously applied options to allow multiple mapped reads for TE expression measurement[38]. The mapped reads were counted for each gene and TE by TEtranscripts version 2.2.1 with mode multi for 1,000 iterations to allow fractional counting of multi-mapped reads[107]. Fold change (FC) of gene expression was analyzed by DESeq2 version 1.42.1[108]. Genes and TEs with no sequencing reads in all samples were excluded from the DESeq2 analysis. Transcripts per million (TPM) were calculated using the established formula[109]. Mean TPM values of three biological replicates for individual genes/TEs were used for the data visualization and statistics.

Global histone H3K27me3 modifications in *V. dahliae* strain JR2 were analyzed using the previous chromatin immunoprecipitation sequencing (ChIP-Seq) data[30] obtained from NCBI SRA under the accession numbers listed in Supplementary Data 14. ChIP-Seq reads were filtered with fastp version 0.19.5[105] and mapped to the JR2 genome masked as previously described[30] using BWA version 0.7.17 with the BWA-MEM algorithm[110]. The mapped reads were sorted and indexed with samtools version 1.10[111] and counted with featureCounts version 2.0.1[112] with the option to count multi-mapped reads fractionally for each genomic compartment over 10 kb windows. The normalized counts per million (CPM) were calculated using the TPM formula[109] by replacing transcripts with bins. The mean normalized

CPM values of two biological replicates for respective bins were used for the data visualization and statistics.

Long-range chromatin interactions in *V. dahliae* strain JR2 were previously identified through chromatin conformation capture (Hi-C)[42]. The previously analyzed data were directly used for visualization with the *Starship* positions.

### Local similarity searches

Local similarity searches were performed with BLAST version 2.15.0+ or 2.16.0+[98]. The nucleotide-to-nucleotide searches were performed with the blastn algorithm with options word size 11 and e-value 0.05. The protein-to-translated nucleotide searches were performed with the tblastn algorithm with an e-value threshold of 0.05. The query *Verticillium* gene and TE sequences were obtained from GenBank and FungiDB under the accession numbers listed in Supplementary Data 15, and genes including introns were used as queries. The sequences and metadata of other virulence-associated proteins were obtained from PHI-base version 4.17[49]. Pezizomycotina genomes were obtained from whole-genome shotgun (WGS) sequences in GenBank under the accession numbers listed in Supplementary Data 9. Fungal taxonomic data were obtained from the National Center for Biotechnology Information (NCBI) Taxonomy database. The coordinates of *Av2* orthologs (Supplementary Data 16) were identified by a blastn search of the Pezizomycotina genomes with *V. dahliae Av2* or the *Av2* ortholog of *Fusarium phyllophilum* (*FpAv2*) under the accession numbers in Supplementary Data 15, followed by the selection of hits with >70% identity and >80% coverage.

Alien index (AI) values were calculated by the formula AI = log((best e-value for ingroup) + e-200) - log((best e-value for outgroup) + e-200)[61] using hits with >50% coverage against *Verticillium* gene/TE queries by the blastn search. The e-value for no hits was set to 1 for queries with hits in either ingroup or outgroup, whereas AI values for queries with no hits in both ingroup and outgroup were not determined. The 332 non-*Verticillium* genomes of the order Glomerellales were used as ingroup, while the 9641 genomes of the remaining 74 orders of Pezizomycotina were used as outgroup (Supplementary Data 9).

### Phylogenetic analyses

*Verticillium* phylogeny was inferred with REALPHY version 1.13[113]. Briefly, genome sequences were fragmented into overlapping 50 bp subsequences and mapped to the JR2 genome with Bowtie version 2.2.5[114]. Based on single-nucleotide polymorphisms in the aligned regions, the maximum likelihood phylogenetic tree was built with PhyML version 3.3.20220408[115].

*Starship* phylogeny was inferred by *k*-mer comparisons with mashtree version 1.4.6 with default *k*-mer size of 21, accuracy option mindepth 0, and bootstrapping for 1,000 iterations[116].

For the phylogenetic analyses of *Av2*, nucleotide sequences of *Av2* orthologs were extracted from the Pezizomycotina genomes at the coordinates listed in Supplementary Data 16 using SeqKit version 2.3.0 subseq[117]. The nucleotide sequences of *Av2* were aligned by

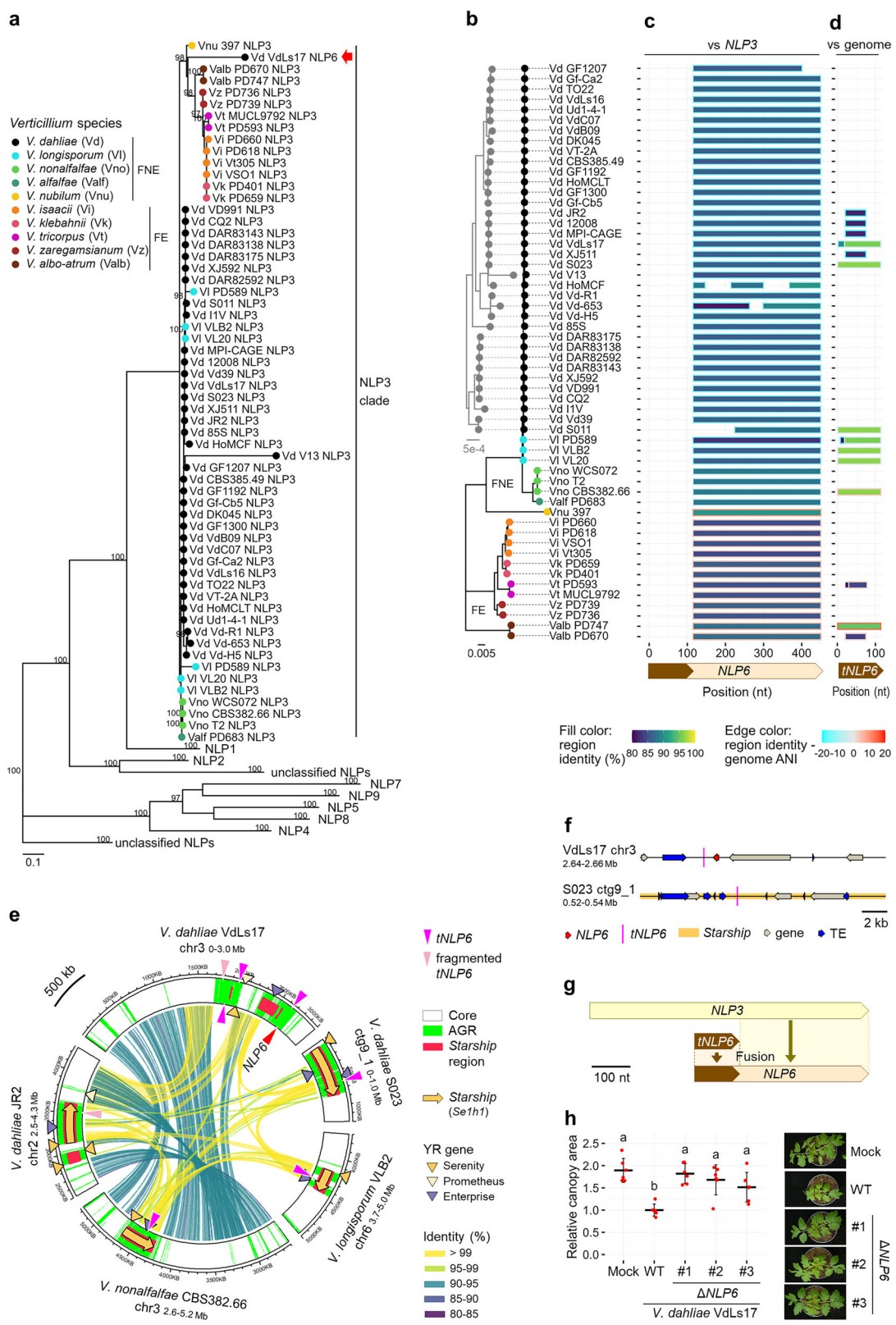

MAFFT versions 7.511 or 7.526 with the L-INS-i method that iteratively refines local alignments[118], followed by removal of ambiguously aligned sites with trimAl version 1.4.rev15 with option strict[119]. After multiple sequence alignments, maximum likelihood phylogeny was inferred by IQ-TREE version 2.0.3[120] with the best substitution model TIM2e + G4 suggested by ModelFinder[121] and the ultrafast bootstrap approximation for 1000 iterations[122].

For the phylogenetic analysis of NLPs, putative *Verticillium* proteins that contain an NPP1 domain[64] were identified based on the homology to reference sequences in the Pfam database[123] (Pfam accession PF05630.16) using HMMER version 3.3.2[96]. Putative secretion signals were identified with SignalP 6.0[124] or SignalP 3.0[125]. The deduced NLP amino acid sequences were aligned and trimmed as described for *Av2*. The phylogeny of NLPs was also inferred as

**Fig. 6 | *Starships* contributed to de novo formation of the virulence gene *NLP6*.**
**a** Phylogeny of necrosis and ethylene-inducing peptide 1 (Nep1)-like proteins (NLPs) in 56 strains across the *Verticillium* genus with all NLP clades collapsed except for the NLP3 clade. The red arrow points to NLP6. Scale bar indicates amino acid substitutions per site. Bootstrap values (>95%) for 1000 iterations are shown at the nodes. Circle colors indicate *Verticillium* spp. from which each NLP homolog was derived. Flavexudans (FE) and Flavnonexudans (FNE) indicate the two clades within the *Verticillium* genus[28]. **b** *Verticillium* phylogeny as described in Fig. 1a. Circle colors indicate *Verticillium* spp. with the same color coding as in (**a**). **c**, **d** Coverage plots for alignments of *NLP6* with *NLP3* genes (**c**) and for alignments of a truncated *NLP6* (*tNLP6*: 1-114 nt of *NLP6*) with genome sequences (**d**). The X-axes represent the nucleotide positions in *NLP6* or *tNLP6* of *V. dahliae* VdLs17. Bar colors indicate the identity of the region and its difference from genome-wide ANI between each *Verticillium* strain and VdLs17. **e** Location of *tNLP6* in *Starship* in three *Verticillium* species. See Fig. 2a legends for the details of symbols. **f** Location of *tNLP6* in non-coding regions of *V. dahliae* strains. **g** Proposed model for *NLP6* evolution. **h** Symptoms of tomato plants inoculated with wild-type (WT) and three independent *NLP6* deletion (Δ) lines of *V. dahliae* strain VdLs17 at 14 days post inoculation. Points indicate relative values for individual plants ($n = 6$), divided by the mean of WT-inoculated plants. Crossbars and error bars indicate mean ± standard deviation. Different letter labels indicate significant differences (two-sided Tukey's test, adjusted $p < 0.05$).

described for *Av2* with the different best substitution model VT + F + I + G4. NLP orthologs were numbered by the monophyletic relationship with NLP1 to NLP9 of *V. dahliae* strain VdLs17[39,63] in the GenBank RefSeq database under the accession numbers listed in Supplementary Data 15.

## Targeted deletion of *NLPs* from the *Verticillium dahliae* genome

To generate *NLP6* and *NLP3* deletion constructs, flanking sequences of its coding sequence were amplified from genomic DNA of *V. dahliae* strains VdLs17 and JR2, respectively, using primers listed in Supplementary Data 17. The amplified products were cloned into pRF-HU2 as described previously[126], and subsequent *Agrobacterium tumefaciens*-mediated transformation of *V. dahliae* was performed as described previously[127]. Transformants were selected on PDA supplemented with cefotaxime (Duchefa, Haarlem, The Netherlands) at 200 µg/ml and hygromycin (Duchefa) at 50 µg/ml, and homologous recombination was PCR-verified. For genetic complementation, the coding sequence of *NLP6* was cloned into the pFBT005 vector as previously described[128], after which the *NLP6* deletion mutants were transformed using the *A. tumefaciens*-mediated transformation method described above.

Pathogenicity assays were performed on ten-day-old tomato seedlings (MoneyMaker) plants using root-dip inoculation as previously described[129]. Disease symptoms were scored up to 14 dpi, pictures were taken, and ImageJ was used to determine canopy areas while fungal colonization was determined with real-time PCR. To this end, stem sections were taken at the height of the first internode, flash-frozen in liquid nitrogen, ground to powder, and genomic DNA was isolated. Real-time PCR was performed with a quantitative PCR core kit for SYBR Green I (Eurogentec, Seraing, Belgium) on an ABI7300 PCR machine (Applied Biosystems, Foster City, CA, U.S.A.). The *V. dahliae* internal transcribed spacer (ITS) levels were used relative to tomato ribulose-1,5-bisphosphate carboxylase/oxygenase (RuBisCO) levels to quantify fungal colonization of tomato plants[36].

## Statistics

Data were analyzed with R version 4.4.2[130] and the R package "tidyverse" version 2.0.0[131]. The normality and homoscedasticity of data were tested by the Shapiro–Wilk test and Bartlett's test, respectively, with the R base package "stats" version 4.3.1[130]. The multiple comparisons of data with non-normal distributions and unequal variances ($p < 0.05$) were performed by Dunn's test with the Bonferroni correction with the R package "dunn.test" version 1.3.6[132]. The multiple comparisons of data with normal distributions and equal variances ($p < 0.05$) were performed by Tukey's test with the R package "multcomp" version 1.4.26[133]. The multiple comparisons of the proportions of two categorical variables were performed by Fisher's test with the Bonferroni correction with the R package "RVAideMemoire" version 0.9-83-11[134]. The exact adjusted $p$-values of multiple comparisons are provided in the Source Data files. The Spearman's rank correlation coefficient was calculated with the R base package "stats" version 4.3.1[130].

## Data visualization

Data were visualized with the R packages. Circular and linear plots of genomic regions were generated with "circlize" version 0.4.16[135] and 'gggenomes' version 1.0.0[136] or "genoPlotR" version 0.8.11[137], respectively. Phylogenetic trees were visualized with "ggtree" version 3.10.1[138]. Other plots were generated with "ggplot2" version 3.5.1[139].

## Reporting summary

Further information on research design is available in the Nature Portfolio Reporting Summary linked to this article.

## Data availability

The genomes assembled in this study and previous studies[57,103] have been submitted to NCBI under the BioProject accession PRJNA1253319. Genome annotation files and genome assemblies with complex gap information[103] have been deposited at Zenodo (https://doi.org/10.5281/zenodo.15450312). Other genome sequence, RNA-Seq, and ChIP-Seq data used in this study are available in the NCBI database under the accession numbers listed in Supplementary Data 1, 9, 10, 14–16. Source data are provided with this paper.

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

## Acknowledgements

Y.S. acknowledges funding through an Overseas Research Fellowship from the Japan Society for the Promotion of Science. B.P.H.J.T. acknowledges funding by the Alexander von Humboldt Foundation in the framework of an Alexander von Humboldt Professorship endowed by the German Federal Ministry of Education and is furthermore supported by the Deutsche Forschungsgemeinschaft (DFG, German Research Foundation) under Germany's Excellence Strategy – EXC 2048/1 – Project ID: 390686111. R.B. and M.H. acknowledge funding by the Research Foundation Flanders (FWO project ID: G004022N). We thank Dr. Edgar A. Chavarro-Carrero for providing sequencing read data of the *V. dahliae* genomes assembled in this study.

## Author contributions

Y.S. and B.P.H.J.T. conceived the project. Y.S., M.F.S., and B.P.H.J.T. designed the analyses. R.B. and M.H. contributed to the discovery of the *NLP6* origin. G.C.M.v.d.B. and P.S. performed fungal gene disruption and pathogenicity assays. Y.S., G.C.M.v.d.B., P.S., and B.P.H.J.T. analyzed the data. Y.S. and B.P.H.J.T. wrote the manuscript with feedback from all authors.

## Funding

## Competing interests

The authors declare no competing interests.
