## [Transparent Peer Review file · Nature Communications]

Starship giant transposons dominate plastic genomic regions in a fungal plant pathogen and drive virulence evolution

Corresponding Author: Professor Bart Thomma

Version 0:

Reviewer comments:

Reviewer #1

(Remarks to the Author)

This study by Sato et al. implicates Starship transposons in the evolution of fungal genomes and reports on possible horizontal gene transfer (HGT) events between *Verticillium dahlia* and *Fusarium* spp. The study concludes that *Verticillium* spp. contains multiple Starships, some of which carry genes typically associated with virulence. The authors show that the Starships found are associated with genomic rearrangements and share traits with plastic AGRs. Finally, they provide evidence for both intra-genus and inter-order HGT events and claimed how a Starship transposition event led to de novo gene birth of a new virulence gene. While we believe this study contributes to the wider fungal genomics community, we cannot recommend it for publication in its current form. We discuss several issues with the paper below that we believe should be addressed prior to publication.

Conclusions overstated given the data provided

Based on the data presented in this study alone, we believe that the manuscript's conclusions are overstated, especially within the abstract and discussion sections. As an example, within the "Cross-order horizontal Starship transfer" section it is stated that the data suggests HGT between *Verticillium* and *Fusarium* (lines 198-199). However, this is portrayed as absolute fact within both the abstract (line 26) and discussion (line 301). We believe the authors should re-address these sections to tone down some of their claims, bringing them more in line with the language used within the results section.

Complex and crowded figures

In its current form the manuscript suffers from overcrowded figures making the results difficult to interpret. Given the broad audience of Nature Communications, simplifying and splitting up some of the figures would be beneficial. In some figures the density makes figure subsections difficult to properly interpret. One example of this is the right-side circular plot of Figure 2A, where more detail is needed to better see the rearrangements at breakpoints and whether they correspond with Starship boundaries.

Another example is Figure 5A and 5B, which are difficult to delineate between. While we understand the reasoning to reshuffle Figure 1A in Figure 5, the accompanying legend should also be shown.

Figure 5B is too dense for readers to fully understand, with a total of 5 dimensions of data to interpret. We recommend the authors to consider breaking figures up or moving sections to supplementary material to improve readability and ability to interpret data.

Lack of quantification and complementation of virulence assays

In its current form, Figure 6H is not suitable for publication. From the subfigure alone, it's difficult to agree with the authors that NLP6 results in a loss-of-virulence phenotype. This is due to the difficulty to visually assess the significance of the phenotype differences between the wildtype infection and the knockout infections. This is especially true of NLP6 #3, where the phenotype looks comparable to the wildtype. However, when considering the quantified data within Figure S8d the significance is clear, and it is easy to agree with authors' conclusions. We recommend that Figure 6H is swapped out for Figure S8d. More seriously the infections assays suffer from a lack of complementation of the knockout lines. Without this we cannot rule out that some aspect of the gene knockout process is responsible for the loss of virulence instead. A

NLP6:NLP6 line should be shown for each replicate of the knockout line.

Confusion surrounding the claimed de novo gene birth of NLP6

In the manuscript's current state, it is difficult to ascertain what the authors conclude is the mechanism responsible for the gene birth of NLP6. While they state that it is formed from a fusion between tNLP6 and 3'-end of NLP3 fusion (line 255-257), it's difficult to conclude what the authors think is responsible for this genomic rearrangement. If the authors are trying to conclude that this rearrangement was caused by Starship transposition, then we would like this section of the results and Figure 6G to be redeveloped to better reflect this hypothesis. Otherwise, if the authors cannot determine the cause of the rearrangement, only that it includes a Starship, we do not agree that this constituted Starship-mediated gene birth.

Overinterpretation of present tyrosine recombinase (YR) genes

One worrying trend of this study is the overinterpretation of the presence of YR genes within the analysed genomes. Sections of the manuscript, such as lines 48-49, makes it seem that the authors correlate all YR genes with Starships. YR domains are not only found in Starship captains, but also Crypton transposons (PMID: 14600222) and certain retrotransposons (PMID: 24086727). Many figures also include YR counts, and it is difficult to determine whether the authors think these are all Starship-related or not. This seems like just an additional dimension of data that is not necessary and might contribute to confusing readers with an assumption that all YRs are Starship-related.

One conclusion that is sort of interesting but completely skipped over is the finding that genes found in Starships are more highly expressed than genes found in other non-starship AGRs, despite there being no difference in heterochromatic mark H3K27. Is this worth discussion how starship might escape genome silencing defences?

Aside from the major comments above, there are sections of the manuscript where long or complex sentences make it difficult to read. We have included examples of these, and other minor details worth addressing, below.

Lines 36-38: while we understand what the authors are trying to convey, in its current form this sentence does not make sense.

Lines 37 and 39: consistency between "giant" TEs' and 'giant TEs.

Lines 77-80: this sentence is lengthy and overcomplicated. For readability we would recommend splitting it up.

Lines 92-95: While it is stated in the methods, explicitly stating the criteria used identify "Starship regions" in the results would provide the reader more clarity on the robustness of these identifications.

Line 94: Disagree that YR genes not found in Starships area robust indication of Starship remnants.

Line 105-106: Difficult to see in this figure due to the small size if the AR regions are at the breakpoints of Starships or not.

Lines 125-128: One sentence paragraph, can this be integrated more thoroughly? Or expanded upon?

Lines 148-151: this sentence is difficult to read, and maybe grammatically incorrect? "whether besides". Please revise.

Lines 139-140: It is not clear if the "segmental duplications" reported here have been described previously for AGRs? Or if they a novel finding of this work?

Line 144-145: If Starships compose ~50% of the length of AGR in JR2 (line 107-108) and 20% of the AGRs in VdLs17 (line 110-111).. then is it a surprise that the two regions have the same features? Does this also contradict the data shown earlier of higher gene expression in Starships compared to non-starship AGRs?

Lines 171-177: This paragraph is too short to full understand the what the authors compared. They refer to two different Starship haplotypes (Se1h1 and Ar1 h1) and then only discuss Ar1h1 insertion in JR2. Could the authors clarify here whether or not this is 5 different independent insertions and/or one insertion followed by movement of the starship.

Reviewer #2

(Remarks to the Author)

Reviewer #3

(Remarks to the Author)

The manuscript by Sato et al describes in great detail the landscape of Starship transposons in across a large suite of genomes from the *Verticillium* genus. There appears to be a truly phenomenal diversity of these elements across this genus and the authors make a really compelling case that these elements have been a real driver of the diversity in the plastic regions of *Verticillium* genomes. The authors go on to demonstrate the likelihood that some horizontal movement of these

transposons has occurred between *Fusarium* spp and *Verticillium* spp and that the Starships are likely to be involved in both the movement of virulence encoding genes and the potential birth of new genes.

The authors use some established bioinformatic tools to characterise these transposons but the transformation of this information into what are truly elegant, and information dense figures is to be commended. I have relatively minor points.

Title: 'a fungal plant pathogen' sound like it is referring to one species but really here multiple pathogens are being examined so this could be changed to a plural. Maybe something like "Giant Starship transposons dominate plastic genomic regions in a genera of fungal plant pathogens and drive virulence evolution"

Line 92: I found the 'starship region' terminology a bit confusing. To me a "starship region" would be the genomic region in which a Starship is found. In line 92 it states these regions are syntenic which by definition means there is homology. But I think based on figure 1e this is more to do with the flanking regions having synteny and maybe a partial Starship. I think these would be better described as "Starship-like regions".

Line 134-135: If I'm interpreting Figure 3 correctly it appears that some of the Ave1 and Av2 homologues aren't in Starships but this sentence suggests they are in all strains. For example Strain GF1207 looks to have Av2 but it is not in a light green square. I'm sure just a minor tweak to this sentence is all that is needed here.

Line 154-155: some of the 'virulence-associated' genes listed here sound more like genes needed for basic fungal growth. E.g. beta-tubulin and even pH-signalling I don't think can really be classed as 'virulence-associated'. These sound more like fairly important genes for fungal life.

Line 204: I don't think the words 'fungal phylogeny' should be used here as the tree in figure 5b is really of two genera that are not sisters (so there is a lot missing in between to really consider this a fungal phylogeny. Perhaps this could be changed to 'clustered according to genera'

Reviewer #4

(Remarks to the Author)

Short summary

Sato et al. investigate the repertoire of giant transposable elements, known as Starships, across 56 genome assemblies of the *Verticillium* genus, which includes the generalist plant pathogen *V. dahliae*. They reveal that Starships are widespread in *Verticillium*, with nearly 60% of assemblies containing at least one element. Analyzing sequence diversity, they highlight an enrichment of structural variations within Starship regions. Furthermore, they demonstrate that these regions exhibit key characteristics of accessory genomic regions (AGRs) in *V. dahliae*, including virulence-associated and in planta-induced genes, elevated TE density and expression, strong H3K27 methylation marks and physically interact in the nucleus.

Exploring the Starship diversity, the authors present evidence of multiple independent horizontal transfers between *Verticillium* species. Expanding their analysis to Pezizomycotina, they identify two instances of cross-order Starship transfer between *Verticillium* and *Fusarium* species. Finally, they investigate the putative virulence gene NLP6, which encodes a Necrosis- and ethylene-inducing peptide 1 (Nep1)-like protein (NLP). They find that NLP6 in *V. dahliae* is located near a Starship element and shares high sequence identity with the C-terminus of NLP3 from *V. nubilum*, supporting the hypothesis that NLP6 originated via horizontal gene transfer followed by Starship-mediated sequence fusion. In a final functional assay, the authors demonstrate that NLP6 deletion impairs *V. dahliae* growth on tomato plants, suggesting a role in virulence.

Short assessment

This study strengthens the growing body of evidence that Starships are key drivers of horizontal gene transfer in Pezizomycotina fungi. By examining a major plant pathogen that affects hundreds of species and lacks a known sexual cycle, this work underscores the crucial role Starships may play in shaping the evolution of clonal fungal pathogens.

Main comments

Overall the manuscript is well written and well articulated, congratulations to all involved. My major comment is regarding the description of the methodology applied and the data availability statement. Most of the methods are currently simply referenced when they should be at least briefly summarised. In addition, most of the assemblies used in the frame of this study, together with their annotations (genes, transposable elements and their consensus sequences) are not publicly available, hindering any form of reproducibility and transparent data access.

Minor comments

line 24 : "Starships are enriched in virulence-associated genes", enrichment here is not directly assessed.

line 92 : here I would provide a bit more detail on what these "Starship regions" are as they are important for the rest of the results section. In addition, it is not always clear further down if referring to Starships only or Starships + Starship regions.

line 115-118 : lower coverage in this case doesn't necessary mean indel as it could simply underly sequence divergence.

line 174-177 : Figure doesn't match the text (Figure 4C-D and S4) - i.e. Ar1h1 present in 10 *V. dahliae* isolates but not present in *V. nonalfafa* (as some *V. dahliae* isolates)

line 181 : hard to visualize "gene gain / losses" on Figure S4D.

line 235 : No conclusive evidence supporting that "Starships mediate de novo gene birth". Indeed, no evidence that the

NLP6 includes the N-term part and is not simply a HGT NLP3 from *V. nubilum*.

line 257 : "a role of NLP3 in fungal virulence could not be demonstrated" the referenced study only show absence of HR in an heterologous expression system, not its role in virulence. Does NLP6 trigger HR in the same system? Also note that no complementation assays were performed here to validate the role of NLP6 in virulence.

Figure 2E : having the distances in kb to match the text could improve readability (y-scale in log10 bp confusing) - similarly, splitting the graphs per variant category could be informative, i.e. is the increase largely due to the fact that there are more TRA and INV (often larger than INDEL / DUPS) at those regions?

Figure 2F : overlapping the two genomic dot plots could better illustrate which regions match one of the other region.

Note that in multiple figures (2,3,4,S4) two Starships often overlap and are annotated in both directions (two arrowheads).

Are these considered to be two distinct Starships or bi-directional elements with captains at both ends?

Table S5 - given the two approaches to define Starship regions I would specify which have the YR or not.

Version 1:

Reviewer comments:

Reviewer #1

(Remarks to the Author)

This is a revised manuscript where the authors have addressed the suggested changes proposed by all reviewers.

There are some new supplementary figures Fig S2 and FigS4b which are new and suggested by reviewers that show in more detail some of the synteny break points between different strains and shuffling of starship sequences. I feel like these new figures are a real improvement and wonder if these could be "upgraded" to the main figure. For example replacing Fig2A with Fig S2. I acknowledge that this is an editorial suggestion.

Reviewer #4

(Remarks to the Author)

Short summary

In their revised manuscript, Sato and co-authors now include a data availability statement with a Zenodo link where all data is accessible, as well as an improved description of the methods. They have also clarified several points in the main text and added five supplementary figures.

While the authors now provide additional evidence supporting a role for NLP6 in *Verticillium dahliae* virulence on tomato, I still find that the contribution of Starships to the formation of NLP6 is overstated.

Specifically, the current data do not provide convincing evidence for a direct role of Starships in either (i) the horizontal gene transfer of NLP3 or (ii) the formation of NLP6. The reported ~100 bp sequence identity is anecdotal at best—especially considering the multi-copy nature of this 5' NLP6 sequence, which could suggest involvement of non-Starship transposon-mediated duplications instead.

REVIEWER COMMENTS

Reviewer #1 (Remarks to the Author):

This study by Sato et al. implicates Starship transposons in the evolution of fungal genomes and reports on possible horizontal gene transfer (HGT) events between Verticillium dahlia and Fusarium spp. The study concludes that Verticillium spp. contains multiple Starships, some of which carry genes typically associated with virulence. The authors show that the Starships found are associated with genomic rearrangements and share traits with plastic AGRs. Finally, they provide evidence for both intra-genus and inter-order HGT events and claimed how a Starship transposition event led to de novo gene birth of a new virulence gene. While we believe this study contributes to the wider fungal genomics community, we cannot recommend it for publication in its current form. We discuss several issues with the paper below that we believe should be addressed prior to publication.

Authors: Thank you very much for your detailed feedback. We have responded to each point below.

Conclusions overstated given the data provided

Based on the data presented in this study alone, we believe that the manuscript's conclusions are overstated, especially within the abstract and discussion sections. As an example, within the "Cross-order horizontal Starship transfer" section it is stated that the data suggests HGT between Verticillium and Fusarium (lines 198-199). However, this is portrayed as absolute fact within both the abstract (line 26) and discussion (line 301). We believe the authors should re-address these sections to tone down some of their claims, bringing them more in line with the language used within the results section.

Authors: We have rephrased several lines in the abstract and in the discussion section to tone down the claims in the revised manuscript.

Complex and crowded figures

In its current form the manuscript suffers from overcrowded figures making the results difficult to interpret. Given the broad audience of Nature Communications, simplifying and splitting up some of the figures would be beneficial. In some figures the density makes figure subsections difficult to properly interpret. One example of this is the right-side circular plot of Figure 2A, where more detail is needed to better see the rearrangements at breakpoints and whether they correspond with Starship boundaries.

Another example is Figure 5A and 5B, which are difficult to delineate between. While we understand the reasoning to reshuffle Figure 1A in Figure 5, the accompanying legend should also be shown.

Figure 5B is too dense for readers to fully understand, with a total of 5 dimensions of data to interpret. We recommend the authors to consider breaking figures up or moving sections to supplementary material to improve readability and ability to interpret data.

Authors: We have added a new Supplementary Figure S2 to better show the rearrangement breakpoints and their correspondence with the Starship boundaries referred to in Figure 2a. We have corrected Figures 6a and 6b to better "delineate between" and reduce "dimensions of data to interpret" (we assume that "Figure 5" was a typographical error in the reviewer's comment that most likely referred to Figure 6 based on the context provided). We have revised the descriptions of revised Figures 6a and 6b to clarify what the two phylogenetic trees show, rather than repeating

the Figure 1a legend. We hope that these changes improve interpretability of the figures for the reviewer.

Lack of quantification and complementation of virulence assays

In its current form, Figure 6H is not suitable for publication. From the subfigure alone, it's difficult to agree with the authors that \square NLP6 results in a loss-of-virulence phenotype. This is due to the difficulty to visually assess the significance of the phenotype differences between the wildtype infection and the knockout infections. This is especially true of \square NLP6 #3, where the phenotype looks comparable to the wildtype. However, when considering the quantified data within Figure S8d the significance is clear, and it is easy to agree with authors' conclusions. We recommend that Figure 6H is swapped out for Figure S8d. More seriously the infections assays suffer from a lack of complementation of the knockout lines. Without this we cannot rule out that some aspect of the gene knockout process is responsible for the loss of virulence instead. A \square NLP6:NLP6 line should be shown for each replicate of the knockout line.

Authors: We have moved the former Supplementary Figure S8d to the revised main Figure 6 as suggested (panel h). We had previously performed pathogenicity assay with complementation lines, which we did not show in the previous version of the manuscript given the wealth of data already in the MS and the large amount of (supplementary) figures, but we have now added these data as revised Supplementary Figure S13 as requested.

Confusion surrounding the claimed de novo gene birth of NLP6

In the manuscript's current state, it is difficult to ascertain what the authors conclude is the mechanism responsible for the gene birth of NLP6. While they state that it is formed from a fusion between tNLP6 and 3'-end of NLP3 fusion (line 255-257), it's difficult to conclude what the authors think is responsible for this genomic rearrangement. If the authors are trying to conclude that this rearrangement was caused by Starship transposition, then we would like this section of the results and Figure 6G to be redeveloped to better reflect this hypothesis. Otherwise, if the authors cannot determine the cause of the rearrangement, only that it includes a Starship, we do not agree that this constituted Starship-mediated gene birth.

Authors: We did not attempt to argue that “this rearrangement was caused by Starship transposition”. Rather, we propose a model that suggests that the birth of NLP6 may have been mediated by genomic rearrangements involving Starships which are significantly more frequently associated with such rearrangements when compared to the other genomic regions. To avoid this misinterpretation, we have now rephrased “Starship activity facilitated the de novo formation” to “a Starship contributed to the de novo formation”. Furthermore, we would like to point out that in such evolutionary analyses it is impossible to “determine the cause of the rearrangement”, simply because it is impossible to go back in time, but we detail, and provide supporting evidence for, the most likely scenario.

Overinterpretation of present tyrosine recombinase (YR) genes

One worrying trend of this study is the overinterpretation of the presence of YR genes within the analysed genomes. Sections of the manuscript, such as lines 48-49, makes it seem that the authors correlate all YR genes with Starships. YR domains are not only found in Starship captains, but also Crypton transposons(PMID: 14600222) and certain retrotransposons (PMID: 24086727). Many figures also include YR counts, and it is difficult to determine whether the authors think these are all

Starship-related or not. This seems like just an additional dimension of data that is not necessary and might contribute to confusing readers with an assumption that all YRs are Starship-related.

Authors: It was not our intention to “correlate all YR genes with Starships” or argue that “all YRs are Starship-related”. Rather, we tried to discuss the possibility that YR genes not currently linked to known *Starships* could be associated with as-yet-unidentified *Starships*, considering the technical limitations of current *Starship* detection methods. This discussion is based on the fact that the YRs analyzed here were identified based on homology to known *Starship* captains rather than to distantly related YRs from other transposons (Gluck-Thaler et al. 2022, PMID: 35588244). To clarify this point, we have now clarified at various instances throughout the manuscript that the YRs discussed in this manuscript are merely those that closely relate to *Starship* captains.

One conclusion that is sort of interesting but completely skipped over is the finding that genes found in Starships are more highly expressed than genes found in other non-starship AGRs, despite there being no difference in heterochromatic mark H3K27. Is this worth discussion how starship might escape genome silencing defences?

Authors: We thank the reviewer for this thoughtful remark, and assume that the term “genes” in this reviewer’s comment refers to TEs, based on the context provided. In response to this reviewer’s question, we have performed additional data analysis and added Figure 3h and Supplementary Figure S5 to the revised manuscript. According to the data presented in these panels, the higher TE expression in *Starship* regions when compared to other genomic compartments is associated with TE sequences rather than with epigenetic features.

Aside from the major comments above, there are sections of the manuscript where long or complex sentences make it difficult to read. We have included examples of these, and other minor details worth addressing, below.

Lines 36-38: while we understand what the authors are trying to convey, in its current form this sentence does not make sense.

Authors: The sentence has been corrected.

Lines 37 and 39: consistency between “giant” TEs’ and ‘giant TEs.

Authors: This has now been corrected.

Lines 77-80: this sentence is lengthy and overcomplicated. For readability we would recommend splitting it up.

Authors: We have split the sentence as recommended.

Lines 92-95: While it is stated in the methods, explicitly stating the criteria used identify “Starship regions” in the results would provide the reader more clarity on the robustness of these identifications.

Authors: A more detailed description has been added.

Line 94: Disagree that YR genes not found in Starships area robust indication of Starship remnants.

Authors: The discussion of *Starship* remnants is not based solely on the presence of YR genes, but rather on the identification of regions syntenic to known *Starships* and their colocalization with genomic rearrangement breakpoints associated with *Starships*, as detailed in the section following this sentence.

Line 105-106: Difficult to see in this figure due to the small size if the AR regions are at the breakpoints of Starships or not.

Authors: We assume that the term “AR regions” in this reviewer’s comment refers to AGRs, based on the context provided. We have now added Supplementary Figure S2 to better show the AGR boundaries and the rearrangement breakpoints. The AGR boundaries do not necessarily correspond exactly to the *Starship* boundaries, as the previous study assigned AGRs and core regions to 10 kb windows rather than to individual bases (Cook et al. 2020, PMID: 33337321).

Lines 125-128: One sentence paragraph, can this be integrated more thoroughly? Or expanded upon?

Authors: The sentence has been merged with the preceding paragraph.

Lines 148-151: this sentence is difficult to read, and maybe grammatically incorrect? “whether besides”. Please revise.

Authors: We have corrected the sentence.

Lines 139-140: It is not clear if the “segmental duplications” reported here have been described previously for AGRs? Or if they a novel finding of this work?

Authors: We have corrected the sentence, and particularly the position of the reference, to clarify that segmental duplications were reported in the previous study.

Line 144-145: If Starships compose ~50% of the length of AGR in JR2 (line 107-108) and 20% of the AGRs in VdLs17 (line 110-111).. then is it a surprise that the two regions have the same features? Does this also contradict the data shown earlier of higher gene expression in Starships compared to non-starship AGRs?

Authors: Although this observation may not be surprising, we consider it essential to assess whether *Starship* regions exhibit traits characteristic of plastic genomic regions to support the key conclusion stated in the manuscript title. Our analyses revealed stronger enrichment of certain AGR traits (e.g., frequent genomic rearrangements and high TE expression) in *Starship* regions. These results are consistent with previous studies but highlight a difference in degree of these traits among regions within AGRs.

Lines 171-177: This paragraph is too short to full understand the what the authors compared. They refer to two different Starship haplotypes (Se1h1 and Ar1 h1) and then only discuss Ar1h1 insertion in JR2. Could the authors clarify here whether or not this is 5 different independent insertions and/or one insertion followed by movement of the starship.

Authors: The synteny plot (revised Supplementary Figure S7c) and a description for *Se1h1* have now been added to the text in the revised manuscript. Our type of analysis will never be able to determine whether an insertion is derived from an external invasion, or from a jump within the genome. From the reviewer comment we infer that the reviewer interprets “insertion” as a “foreign invasion”. Therefore, we now changed the term “insertion” to “movement” as a more neutral term and embrace both possibilities.

Reviewer #2 (Remarks to the Author):

Reviewer #3 (Remarks to the Author):

The manuscript by Sato et al describes in great detail the landscape of Starship transposons in across a large suite of genomes from the Verticillium genus. There appears to be a truly phenomenal diversity of these elements across this genus and the authors make a really compelling case that these elements have been a real driver of the diversity in the plastic regions of Verticillium genomes. The authors go on to demonstrate the likelihood that some horizontal movement of these transposons has occurred between Fusarium spp and Verticillium spp and that the Starships are likely to be involved in both the movement of virulence encoding genes and the potential birth of new genes.

The authors use some established bioinformatic tools to characterise these transposons but the transformation of this information into what are truly elegant, and information dense figures is to be commended. I have relatively minor points.

Authors: Thank you very much for your praise and feedback.

Title: ‘a fungal plant pathogen’ sound like it is referring to one species but really here multiple pathogens are being examined so this could be changed to a plural. Maybe something like “Giant Starship transposons dominate plastic genomic regions in a genera of fungal plant pathogens and drive virulence evolution”

Authors: Although we appreciate the suggestion, we would like to prevent over-statement of our conclusions, as pointed out by referee 1. A particular reason to focus mainly on *V. dahliae*, and therefore refer to a single species only, is that our determination of plastic genomic regions (AGRs) is rather crude and rudimentary, with much less detail. Thus, we are cautious generalizing our conclusions at present decided to stay with the original title.

Line 92: I found the ‘starship region’ terminology a bit confusing. To me a “starship region” would be the genomic region in which a Starship is found. In line 92 it states these regions are syntenic which by definition means there is homology. But I think based on figure 1e this is more to do with the flanking regions having synteny and maybe a partial Starship. I think these would be better described as “Starship-like regions”.

Authors: We considered the possibility that these “Starship regions” may include Starships overlooked by the current analysis due to the technical limitations discussed in the manuscript. In this light, we feel that the term “Starship-like region” may also be confusing and not accurately reflect their true nature. To maintain consistency in our definition for subsequent analyses comparing Starship regions (i.e., Starships and their syntenic regions) with other genomic regions, we have retained the term “Starship regions”.

Line 134-135: If I’m interpreting Figure 3 correctly it appears that some of the Ave1 and Av2 homologues aren’t in Starships but this sentence suggests they are in all strains. For example Strain GF1207 looks to have Av2 but it is not in a light green square. I’m sure just a minor tweak to this sentence is all that is needed here.

Authors: In Figure 3b, only the colored strains contain Starships that were confidently identified using Starfish (Gluck-Thaler and Vogan. 2024, PMID: 38686785). In the other strains, Ave1 and

Av2 homologs were found in genomic regions syntenic to these *Starships*. We have revised the figure legend to clarify this.

Line 154-155: some of the 'virulence-associated' genes listed here sound more like genes needed for basic fungal growth. E.g. beta-tubulin and even pH-signalling I don't think can really be classed as 'virulence-associated'. These sound more like fairly important genes for fungal life.

Authors: We understand the comment of the reviewer, but these are genes that have been described in studies by others and were shown, in these studies, to contribute to virulence and labelled as virulence genes. Therefore, we have revised the sentence to clarify that these genes have been experimentally shown to contribute to virulence.

Line 204: I don't think the words 'fungal phylogeny' should be used here as the tree in figure 5b is really of two genera that are not sisters (so there is a lot missing in between to really consider this a fungal phylogeny. Perhaps this could be changed to 'clustered according to genera'

Authors: We have revised the sentence as suggested.

Reviewer #4 (Remarks to the Author):

Short summary

Sato et al. investigate the repertoire of giant transposable elements, known as Starships, across 56 genome assemblies of the Verticillium genus, which includes the generalist plant pathogen V. dahliae. They reveal that Starships are widespread in Verticillium, with nearly 60% of assemblies containing at least one element. Analyzing sequence diversity, they highlight an enrichment of structural variations within Starship regions. Furthermore, they demonstrate that these regions exhibit key characteristics of accessory genomic regions (AGRs) in V. dahliae, including virulence-associated and in planta-induced genes, elevated TE density and expression, strong H3K27 methylation marks and physically interact in the nucleus.

Exploring the Starship diversity, the authors present evidence of multiple independent horizontal transfers between Verticillium species. Expanding their analysis to Pezizomycotina, they identify two instances of cross-order Starship transfer between Verticillium and Fusarium species. Finally, they investigate the putative virulence gene NLP6, which encodes a Necrosis- and ethylene-inducing peptide 1 (Nep1)-like protein (NLP). They find that NLP6 in V. dahliae is located near a Starship element and shares high sequence identity with the C-terminus of NLP3 from V. nubilum, supporting the hypothesis that NLP6 originated via horizontal gene transfer followed by Starship-mediated sequence fusion. In a final functional assay, the authors demonstrate that NLP6 deletion impairs V. dahliae growth on tomato plants, suggesting a role in virulence.

Short assessment

This study strengthens the growing body of evidence that Starships are key drivers of horizontal gene transfer in Pezizomycotina fungi. By examining a major plant pathogen that affects hundreds of species and lacks a known sexual cycle, this work underscores the crucial role Starships may play in shaping the evolution of clonal fungal pathogens.

Authors: Thank you very much for your evaluation and feedback. We have responded to your comments as follows:

Main comments

Overall the manuscript is well written and well articulated, congratulations to all involved. My major comment is regarding the description of the methodology applied and the data availability statement. Most of the methods are currently simply referenced when they should be at least briefly summarised. In addition, most of the assemblies used in the frame of this study, together with their annotations (genes, transposable elements and their consensus sequences) are not publicly available, hindering any form of reproducibility and transparent data access.

Authors: We have added short summaries of methods where these had been missing. We have deposited the genome assemblies and annotations in publicly available databases (see data availability statement). As we will provide detailed source data for each figure, redundant Supplementary Tables in the initial submission have been removed.

Minor comments

line 24 : "Starships are enriched in virulence-associated genes", enrichment here is not directly assessed.

Authors: The description has been revised to: "*Starships* carry multiple virulence-associated genes".

line 92 : here I would provide a bit more detail on what these "Starship regions" are as they are important for the rest of the results section. In addition, it is not always clear further down if referring to Starships only or Starships + Starship regions.

Authors: A more detailed description of *Starship* regions has been added to the Results section. We have emphasized that 'Starships' refer to elements currently identified by Starfish and 'Starship regions' include both *Starships* and their syntenic regions.

line 115-118 : lower coverage in this case doesn't necessary mean indel as it could simply underly sequence divergence.

Authors: We have added the possibility of sequence divergence in the revised manuscript.

line 174-177 : Figure doesn't match the text (Figure 4C-D and S4) - i.e. Ar1h1 present in 10 V. dahliae isolates but not present in V. nonalfafa (as some V. dahliae isolates)

Authors: In Figures 4C-D and S4, we had shown only *Starships* confidently identified by Starfish. In *V. nonalfalfae*, *Ar1h1* *Starships* could not be identified by Starfish due to the limited availability of *V. nonalfalfae* genomes, while the regions syntenic to the nearly full-length *Ar1h1* *Starships* were identified by synteny searches. We clarified these points in the revised manuscript and in the legend of revised Supplementary Figure S7b.

line 181 : hard to visualize "gene gain / losses" on Figure S4D.

Authors: Additional figures have been generated. Please see the revised Supplementary Figure S8c that show gain or loss of cargo TEs.

line 235 : No conclusive evidence supporting that "Starships mediate de novo gene birth". Indeed, no evidence that the NLP6 includes the N-term part and is not simply a HGT NLP3 from V. nubilum.

Authors: We have proposed a model based on the evolutionary analyses, as the gene birth process is impossible to observe directly. The N-terminal part of NLP6 showed no sequence homology to the N-terminal parts of the known NLP3 orthologs in *V. nubilum* and other *Verticillium* species, but was nearly identical to the *Starship* non-coding element in nucleotide sequence. Therefore, we have proposed the fusion model involving an NLP3 ortholog and the *Starship* non-coding element for the origin of NLP6, rather than sequence divergence of an NLP3 ortholog.

line 257 : "a role of NLP3 in fungal virulence could not be demonstrated" the referenced study only show absence of HR in an heterologous expression system, not its role in virulence. Does NLP6 trigger HR in the same system? Also note that no complementation assays were performed here to validate the role of NLP6 in virulence.

Authors: In the cited study, the lack of *NLP3* contribution towards virulence had been experimentally observed, but the data had not been included. We have now added this data in

Supplementary Figure S13, and added the PhD student who was involved in the work at that time as a coauthor to the current study. It is important to note that *NLP6* did not induce cell death in the same system (Santhanam et al. 2013, PMID: 23051172). Moreover, we have now added the analysis and description of the *NLP6* complementation lines, as similarly requested by reviewer 1, in the revised Supplementary Figure S13.

Figure 2E : having the distances in kb to match the text could improve readability (y-scale in log10 bp confusing) - similarly, splitting the graphs per variant category could be informative, i.e. is the increase largely due to the fact that there are more TRA and INV (often larger than INDEL / DUPS) at those regions?

Authors: We have modified the scale of Figure 2e and have added the revised Supplementary Figure S4a that shows “the graphs per variant category”. *Starship* regions on average have a larger unaligned part than the core genomic regions for every variant.

Figure 2F : overlapping the two genomic dot plots could better illustrate which regions match one of the other region.

Authors: We have added the superimposed plots as Supplementary Figure S4b.

*Note that in multiple figures (2,3,4,S4) two *Starships* often overlap and are annotated in both directions (two arrowheads). Are these considered to be two distinct *Starships* or bi-directional elements with captains at both ends?*

Authors: These elements indicate the “*bi-directional elements with captains at both ends*” (single *Starships* whose direction could not be determined due to the presence of captains on both ends, likely resulting from nested *Starship* insertions). To aid understanding, we have added this explanation.

*Table S5 - given the two approaches to define *Starship* regions I would specify which have the YR or not.*

Authors: We have added the subdivided coordinates of *Starship* regions identified by each approach. See revised Tables S12 and S13.

POINT-BY-POINT RESPONSE TO THE REVIEWERS' COMMENTS IN THE SECOND REVISION

Reviewer #1 (Remarks to the Author):

There are some new supplementary figures Fig S2 and FigS4b which are new and suggested by reviewers that show in more detail some of the synteny break points between different strains and shuffling of starship sequences. I feel like these new figures are a real improvement and wonder if these could be "upgraded" to the main figure. For example replacing Fig2A with Fig S2. I acknowledge that this is an editorial suggestion.

Authors: Thank you very much for your suggestion. Although we see the point that the details provided in the new supplementary figure panels are worthwhile, we prefer to keep the figures as they were, given that the present main figures provide a global (over)view and collate more information than Figs. S2 and S4b. However, we have added the citations for Supplementary Figures S2 and S4b to the legend of Fig. 2A, specifically directing the readers to these figures for more detailed information.

Reviewer #4 (Remarks to the Author):

While the authors now provide additional evidence supporting a role for NLP6 in *Verticillium dahliae* virulence on tomato, I still find that the contribution of Starships to the formation of NLP6 is overstated.

Specifically, the current data do not provide convincing evidence for a direct role of Starships in either (i) the horizontal gene transfer of NLP3 or (ii) the formation of NLP6. The reported ~100 bp sequence identity is anecdotal at best—especially considering the multi-copy nature of this 5' NLP6 sequence, which could suggest involvement of non-Starship transposon-mediated duplications instead

Authors: The suggested involvement of non-Starship transposon-mediated duplications instead the mechanism proposed in our manuscript seems rather unlikely, given that no non-*Starship* transposons directly flanking or carrying either *NLP6* or *tNLP6* have yet been detected, as shown in Fig. 6f. Thus, we deem our proposed Starship-mediated hypothesis more likely, given all the evidence presented in the manuscript. However, we see the point of the reviewer and have toned down the conclusions regarding the evolution of *NLP6* in the manuscript. Whereas we previously stated in the abstract and the discussion that our data “show”, we now state the data “suggest”.